# MAXIMUM ENTROPY POPULATION BASED TRAINING FOR ZERO-SHOT HUMAN-AI COORDINATION

## ABSTRACT

An AI agent should be able to coordinate with humans to solve tasks. We consider the problem of training a Reinforcement Learning (RL) agent without using any human data, i.e., in a zero-shot setting, to make it capable of collaborating with humans. Standard RL agents learn through self-play. Unfortunately, these agents only know how to collaborate with themselves and normally do not perform well with unseen partners, such as humans. The methodology of how to train a robust agent in a zero-shot fashion is still subject to research. Motivated by maximum entropy RL, we derive a centralized population entropy objective to facilitate learning of a diverse population of agents, which is later used to train a robust AI agent to collaborate with unseen partners. The proposed method shows its effectiveness compared to baseline methods, including self-play PPO, the standard Population-Based Training (PBT), and trajectory diversity-based PBT, in the Overcooked game environment. We also conduct online experiments with real humans and further demonstrate the efficacy of the method in the real world.

## 1 INTRODUCTION

Deep Reinforcement Learning (RL) has gained many successes against humans in competitive games, such as Go (Silver et al., 2017), Dota (OpenAI, 2019), and StarCraft (Vinyals et al., 2019a). However, it still remains a challenge to build AI agents that can coordinate and collaborate with humans that the agents have not seen during training (Kleiman-Weiner et al., 2016; Lerer & Peysakhovich, 2017; Carroll et al., 2019; Shum et al., 2019; Hu et al., 2020; Knott et al., 2021). Zero-shot human-AI coordination is particularly important in real-world applications, such as cooperative games (Carroll et al., 2019), communicative agents (Foerster et al., 2016), self-driving vehicles (Resnick et al., 2018), and assistant robots (Andrychowicz et al., 2018). Ultimately, we want to build AI systems that can assist humans and augmenting our capabilities (Engelbart, 1962; Carter & Nielsen, 2017) to make our life better.

The mainstream method for building state-of-the-art AI agents is through self-play reinforcement learning (Tesauro, 1994; Silver et al., 2017). Self-play-trained agents are very specialized, and therefore suffered significantly from distributional shift when paired with humans. For example, in the Overcooked game, the self-play-trained agents only use a specific pot and ignore the others. However, humans use all pots. The AI agent ends up waiting unproductively for the human to deliver a soup from the specific pot, while the human has instead decided to fill up the other pots (Carroll et al., 2019). Since the AI agent has only encountered its own policy during training, it undergoes a distributional shift when it is paired with human players.

We explore how to design a robust and efficient approach to train agents for zero-shot human-AI coordination. To that end, we draw on the advances in maximum entropy RL (Haarnoja et al., 2018b), diversity (Eysenbach et al., 2019), and Multi-agent RL (Lowe et al., 2017; Foerster et al., 2018). First, maximum entropy RL augments the standard reward function with an entropy maximization term and provides a substantial improvement in exploration and robustness (Ziebart et al., 2008; Toussaint, 2009; Ziebart, 2010; Rawlik et al., 2013; Fox et al., 2015; Haarnoja et al., 2017; 2018b; Zhao et al., 2019). For a population of agents, we use the maximum entropy bonus to encourage individual diversity, i.e., each agent's policy to be diverse and exploratory. Secondly, to acquire diverse and distinguishable behaviors (Eysenbach et al., 2019), we further utilize the sum of cross-entropy terms across all agent pairs in the population to encourage pairwise diversity. We define the

combination of individual diversity and pairwise diversity as population diversity. Thirdly, analog to multi-agent RL (Lowe et al., 2017; Foerster et al., 2018), the population diversity is calculated in a centralized fashion. Each agent in the population is rewarded to maximize the centralized population diversity. Subsequently, we derive a safe and computationally efficient surrogate objective, i.e., population entropy, which is proven to be a lower bound of the original population diversity objective. The population entropy is defined as the entropy of the averaged action probability distribution conditioned on the state across all agents in the population. Eventually, we train a new agent to collaborate with each agent in the diversified population using prioritized sampling of agents policies from the population based on the learning progress (Schaul et al., 2016; Vinyals et al., 2019a;b; Han et al., 2020). The newly trained AI agent encounters a diverse set of strategies and is in general more robust to human behaviors (Pan & Yang, 2009; Tobin et al., 2017; Andrychowicz et al., 2018). We evaluate the proposed Maximum Entropy Population-based training (MEP) framework in a testbed based on the popular Overcooked game (Ghost Town Games, 2016).

Our contributions are three-fold. First, based on the novel population diversity objective that considers both agents' individual diversity and pairwise diversity, we derive a safe and computationally efficient surrogate objective, i.e., the population entropy. Secondly, we develop the MEP framework, which comprises training a diversified population and using this population to train a robust AI agent. Last but not least, in the experiments, we show MEP's superior performance, by comparing it with state-of-the-art baseline methods. To further verify the improvements, we conduct online experiments with real humans.

## 2 PRELIMINARIES

**Markov Decision Process:** A two-player Markov Decision Process (MDP) is defined by a tuple $\mathcal{M} = \langle \mathcal{S}, \{\mathcal{A}^{(i)}\}, \mathcal{P}, \gamma, R \rangle$ (Boutilier, 1996), where $\mathcal{S}$ is a set of states; $\mathcal{A}^{(i)}$ is a set of the $i$-th agent's actions, where $i \in [1, 2]$; $\mathcal{P}$ is a transition function that maps the current state and all agents' actions to the next state; $\gamma$ is the discount factor; $R$ is a reward function. The $i$-th agent's policy is $\pi^{(i)}$. A trajectory is denoted by $\tau$. The shared objective is to maximize the expected sum rewards, which is $\mathbb{E}_\tau \left[ \sum_t R(s_t, a_t) \right]$, where $a_t = (a_t^{(1)}, a_t^{(2)})$. We can extend the objective to infinite horizon problems by the discount factor $\gamma$ to ensure that the sum of expected rewards is finite.

**AI Agent, Population, and Human:** Throughout this paper, we use the phrase *AI agent* to explicitly mean the agent that plays the AI role in human-AI coordination. The *population* of agents is used to train the AI agent to make it capable of cooperating with different partner agents. The *human* policy is represented as $\pi^{(H)}$ and a model of the human policy is $\hat{\pi}^{(H)}$. The AI agent is denoted as $\pi^{(A)}$.

**Environment:** We use the Overcooked environment (Carroll et al., 2019) as the human-AI coordination testbed, see Figure 2. In the Overcooked game, to have a high score, it naturally requires coordination and collaboration between the two players. The players are tasked to cook soups.

**Maximum Entropy RL:** Standard RL maximizes the expected sum of rewards $\mathbb{E}_\tau \left[ \sum_t R(s_t, a_t) \right]$. At the beginning of learning, almost all actions have equal probability. After some training, some actions have a higher probability in the direction of accumulating more rewards. Subsequently, the entropy of the policy is reduced over time during training (Mnih et al., 2016). Maximum entropy RL augments the standard RL objective with a maximum entropy term (Ziebart, 2010; Haarnoja et al., 2018b), which gives a reward to the agent if it selects the non-dominate actions during training, and the higher the reward favors more exploration. The maximum entropy RL objective is defined as:

$$J(\pi) = \mathbb{E}_\tau \left[ \sum_t R(s_t, a_t) + \alpha \mathcal{H}(\pi(\,\cdot\,|s_t)) \right]. \tag{1}$$

The parameter $\alpha$ adjusts the relative importance of the entropy bonus against the reward and controls the stochasticity of the optimal policy. The maximum entropy RL objective has a number of advantages. First, the policy is incentivized to explore more widely. Prior works have demonstrated improved exploration using the maximum entropy RL objective (Haarnoja et al., 2017; Schulman et al., 2017a). Secondly, the policy can capture multiple modes of optimal behaviors. In situations where multiple actions are equally important, the policy will give equal probability mass to those actions. Lastly, recent works have shown improved robustness of the policy trained with maximum entropy RL (Haarnoja et al., 2018a; 2019).

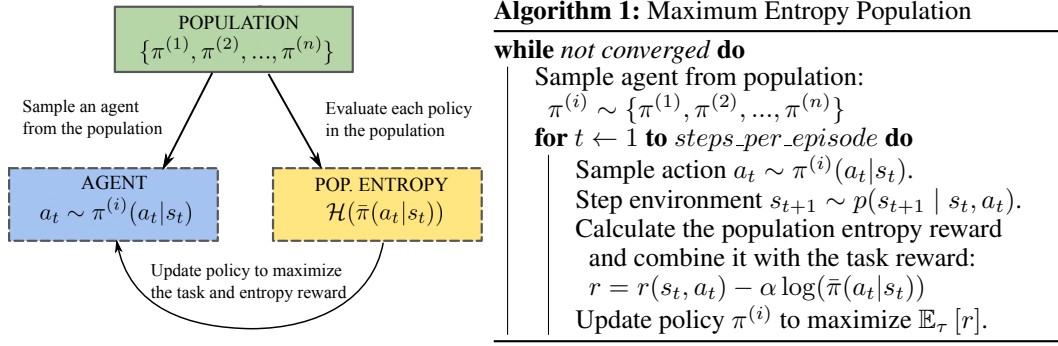

Figure 1: **Maximum Entropy Population**: We train each agent in the population to maximize its task reward as well as the population entropy reward to attain a maximum entropy population.

## 3 METHOD

In this section, we first define the population diversity objective, which includes the policy entropy and the pairwise difference among policies. Secondly, we derive a safe and computationally efficient surrogate objective, i.e., the population entropy, for optimization. Thirdly, we illustrate the MEP framework, which comprises two parts: training a maximum entropy population and training a robust AI agent via prioritized sampling using the population.

### 3.1 POPULATION DIVERSITY

Motivated by maximum entropy RL, we want to make the policy population exploratory and diverse. First, by utilizing the maximum entropy bonus, we encourage each policy itself to be diverse and multi-modal. Secondly, to encourage the policies in the population, $\{\pi^{(1)}, \pi^{(2)}, ..., \pi^{(n)}\}$, to be complementary and mutually different, we naturally utilize the cross entropy objective (Murphy, 2012). Formally, we define the Population Diversity (PD) as the combination of the sum of the entropy of each agent's policy and the sum of the Cross Entropy (CE) between each pair of agents in the population. Mathematically,

$$\text{PD}(\{\pi^{(1)}, \pi^{(2)}, ..., \pi^{(n)}\}, s_t) := \sum_{i=1}^{n} \sum_{j \neq i}^{n} \text{CE}(\pi^{(i)}(\cdot | s_t), \pi^{(j)}(\cdot | s_t)) + \sum_{i=1}^{n} \mathcal{H}(\pi^{(i)}(\cdot | s_t)), \quad (2)$$

where cross entropy (CE) and entropy ($\mathcal{H}$) are defined as follows:

$$\text{CE}(\pi^{(i)}(\cdot | s_t), \pi^{(j)}(\cdot | s_t)) = -\sum_{a \in \mathcal{A}} \pi^{(i)}(a_t | s_t) \log \pi^{(j)}(a_t | s_t), \quad (3)$$

$$\mathcal{H}(\pi^{(i)}(\cdot | s_t)) = -\sum_{a \in \mathcal{A}} \pi^{(i)}(a_t | s_t) \log \pi^{(i)}(a_t | s_t). \quad (4)$$

This objective not only captures single agent's diversity but also encourage multiple agents to be far from each other. However, the population diversity objective is not safe to be maximized as a reward function, since cross entropy is unbounded. Besides, evaluating the population diversity objective has a quadratic runtime complexity of $O(n^2)$, where $n$ is the population size.

### 3.2 POPULATION ENTROPY

To improve the stability and the runtime complexity, we derive a safe and efficient surrogate objective for optimization, which is named the Population Entropy (PE). Population entropy is defined as the entropy of the mean of all policies in the population. Mathematically,

$$\text{PE}(\{\pi^{(1)}, \pi^{(2)}, ..., \pi^{(n)}\}, s_t) := \mathcal{H}(\bar{\pi}(\cdot | s_t)), \text{ where } \bar{\pi}(a_t | s_t) := \frac{1}{n} \sum_{i=1}^{n} \pi^{(i)}(a_t | s_t). \quad (5)$$

The population entropy serves as a lower bound of the population diversity objective.

**Theorem 1.** *Let the population diversity be defined as Equation (2). Let the population entropy be defined as Equation (5). Then, we have*

$$PD(\{\pi^{(1)}, \pi^{(2)}, ..., \pi^{(n)}\}, s_t) \geq n^2 PE(\{\pi^{(1)}, \pi^{(2)}, ..., \pi^{(n)}\}, s_t), \qquad (6)$$

*where $n$ is the population size. Proof. See Appendix A.* □

Compared to the population diversity objective, the population entropy objective, $\mathcal{H}(\bar{\pi}(\cdot|s_t))$, is safe to maximize and it has only a linear runtime complexity $O(n)$. Therefore, we use the derived population entropy objective for optimization. Taking a closer look at the population entropy objective, we find that it can be written into the Jensen-Shannon Divergence (JSD) and entropy form as follows:

$$\mathcal{H}(\bar{\pi}(\cdot|s_t)) = \text{JSD}(\pi^{(1)}(\cdot|s_t), ..., \pi^{(n)}(\cdot|s_t)) + \frac{1}{n}\sum_{i=1}^{n}\mathcal{H}(\pi^{(i)}(\cdot|s_t)), \qquad (7)$$

A step-by-step derivation is in Appendix B. From Equation (7), we can see that when we maximize the population entropy objective, it attempts to push each policy away from each other via the JSD term in Equation 7, as well as increase each policy's entropy via the entropy term in Equation 7.

### 3.3 TRAINING A MAXIMUM ENTROPY POPULATION

We want to train a population of agents, who can play well with themselves. In the meantime, we also want their strategies to be different from each other. Subsequently, we define the reward function for MEP as follows:

$$J(\pi) = \mathbb{E}_\tau\left[\sum_t R(s_t, a_t) + \alpha\mathcal{H}(\bar{\pi}(\cdot|s_t))\right], \text{ where } \bar{\pi}(\cdot|s_t) := \frac{1}{n}\sum_{i=1}^{n}\pi^{(i)}(\cdot|s_t). \qquad (8)$$

Equation (8) tells us that for each agent in the population, the agent is trained to maximize its task reward as well as the centralized population entropy objective. The task reward is related to the agent and its partner agent, which is a copied version of itself playing the partner role in our case. When the centralized population entropy reward is calculated, it considers all the agents in the population. We summarize the method of training a maximum entropy population in Algorithm 1 and Figure 1.

After having the maximum entropy population, we utilize this diverse set of agents to train the AI agent to be ready to pair with human players. The intuition behind MEP is that the AI agent should be more robust when paired with a group of diversified partners during training than trained only via self-play. In the extreme case, when the AI agent can coordinate well with an infinite set of different partners, it can also collaborate well with humans. In a more realistic sense, the more diverse the population is, it is more likely to cover most of the human behaviors in the training set. Subsequently, the final AI agent should be less "panicked" when facing "abnormal" human actions.

### 3.4 TRAINING THE AGENT VIA PRIORITIZED SAMPLING

Considering that playing with different partner agents in the population, the AI agent needs different amounts of training time to learn to coordinate with them. We propose to use prioritized sampling based on the learning progress (Vinyals et al., 2019b), i.e., the expected accumulated reward, to adjust the frequency of each partner agent in the population to occur during training. Mathematically, the probability of the $i$-th agent to be sampled is:

$$p(\pi^{(i)}) = \frac{\text{rank}\left(1/\mathbb{E}_\tau\left[\sum_t R(s_t, a_t^{(A)}, a_t^{(i)})\right]\right)^\beta}{\sum_{j=1}^{n}\text{rank}\left(1/\mathbb{E}_\tau\left[\sum_t R(s_t, a_t^{(A)}, a_t^{(j)})\right]\right)^\beta}, \qquad (9)$$

where the superscript $(A)$ refers to the AI agent that we train for coordinating with humans; $n$ is the population size; $\text{rank}(\cdot)$ is the ranking function ranging from 1 to $n$; $\beta$ is a hyper-parameter for adjusting the strength of the prioritization. We assign a higher priority to the agents that are relatively harder to collaborate with. In the extreme case, at each optimization step, if we always choose the hardest agent in the population to train the AI agent, then we optimize a performance lower bound of the cooperation between the AI agent and any agent in the population. Mathematically,

$$\pi^{(A)} = \arg\max\min_{i \in \{1,...,n\}} J(\pi^{(A)}, \pi^{(i)}), \qquad (10)$$

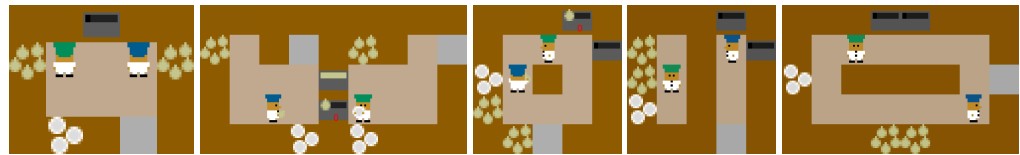

Figure 2: **Overcooked environment**: From left to right, the layouts are *Cramped Room*, *Asymmetric Advantages*, *Coordination Ring*, *Forced Coordination*, and *Counter Circuit*.

where $J(\pi^{(A)}, \pi^{(i)})$ denotes the expected sum reward achieved by $\pi^{(A)}$ and $\pi^{(i)}$ collaborating with each other. With prioritized sampling, we make the collaboration between the AI agent and any agent in the population as good as possible. While uniform sampling does not provide any guarantee on the worst case. For more detail on the performance lower bound, i.e., Equation (10), see Lemma 4 in Appendix C. Furthermore, we derive the performance connection between two pairs of agents, i.e., $(\pi^{(A)}, \pi^{(i)})$ and $(\pi^{(A)}, \pi^{(j)})$, when the partner agent $\pi^{(i)}$ in the first pair is $\epsilon$-close (Ko, 2006) to the other partner agent $\pi^{(j)}$ in the second pair, see Lemma 5 in Appendix D. Based on Lemma 5, if the population we used for training is diverse and representative enough, then we can find an agent that is $\epsilon$-close to the human player's policy and have a performance lower bound of human-AI coordination. In this case, prioritized sampling optimizes not only the performance lower bound of the AI agent and the population, see Equation (10), but also the performance lower bound between the human player and the AI agent, see Corollary 1 in Appendix D.

## 4 EXPERIMENTS

**Environment:** To evaluate the proposed method, we first use a toy environment, i.e., the matrix game (Lupu et al., 2021), see Figure 3, and then use the Overcooked environment (Carroll et al., 2019), see Figure 2. The Overcooked game naturally requires human-AI coordination to achieve a high score. The players are tasked to cook the onion soups as fast as possible. The relevant objects are onions, plates, and soups. Players are required to place 3 onions in a pot, cook them for 20 timesteps, put the cooked soup in a plate, and serve the soup. Afterwards, the players receive a reward of 20. The six actions are up, down, left, right, noop, and interact. There are five different layouts, see Figure 2. Each layout has a different challenge. For example, in *Asymmetric Advantages*, good players should discover their advantages and play to their strengths. The player in the left has the advantage to deliver the soup. The player in the right is closer to the onions.

**Experiments:** First, we train the population using the population entropy reward and investigate the effect of the entropy weight $\alpha$. Secondly, we use the learned maximum entropy population to train the AI agent with the learning progress-based prioritized sampling and report the performance. In an ablation study, we show the effectiveness of both population entropy and prioritized sampling. We compare our method with other methods, including Self-Play (SP) Proximal Policy Optimization (PPO) (Schulman et al., 2017b; Carroll et al., 2019), Population Based Training (PBT) (Jaderberg et al., 2017; Carroll et al., 2019), and Trajectory Diversity (TrajeDi)-based PBT (Lupu et al., 2021). To test the methods, we use the protocol proposed by Carroll et al. (2019), in which a human proxy model, $H_{Proxy}$, is used for evaluation. The human proxy model is trained through behavior cloning (Bain & Sammut, 1999) on the collected human data. Furthermore, we conduct a user study using Amazon Mechanical Turk (AMT), in which we deploy our models through web interfaces and let real human players play with the AI agents. The experimental details are shown in Appendix E. Our code is available as supplementary material.

**Question 1.** *How does MEP perform in the toy environment, i.e., the matrix game?*

In the single-step collaborative matrix game (Lupu et al., 2021), player 1 must select a row while player 2 chooses a column independently. Both agents get the reward associated with the intersection of their choices at the end of the game. We use the same evaluation protocol as proposed by Lupu et al. (2021). From Figure 3, we can see that MEP converges faster than TrajeDi. We did an extensive hyper-parameter search for TrajeDi, as shown in Figure 7 in Appendix F.

**Question 2.** *Does the population entropy reward increase the entropy of the population?*

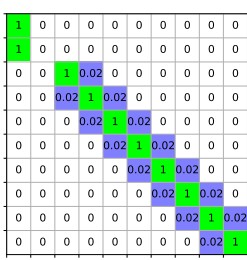 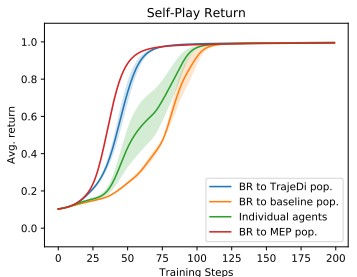 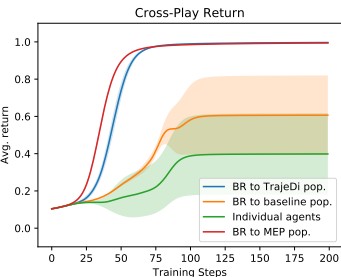

Figure 3: **Performance comparison**: Train and test performances on the matrix game. Shown are the results for Best Responses (BRs) to MEP agents, BRs to TrajeDi populations, BRs to baseline populations, and individual agents. MEP allows faster learning compared to TrajeDi and others.

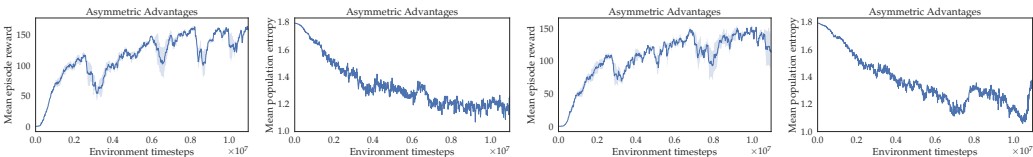

Figure 4: **Mean episode reward and population entropy learning curve**: The left two plots are the mean episode reward and population entropy with entropy reward weight $\alpha = 0$ in the Asymmetric Advantage layout. The right two plots show the quantities with $\alpha = 0.01$ in the same layout.

To verify if the population entropy reward indeed increases the entropy of the population, we monitor the population entropy during training. The learning curve of the mean episode reward of the population and its corresponding population entropy is shown in Figure 4. This figure shows that as the training starts, the reward increases, and the population entropy decreases. Comparing the left two figures with $\alpha = 0$ and the right two figures with $\alpha = 0.01$, we can see that with the population entropy reward, the population entropy indeed converges to a higher value. We also try different values of $\alpha$ and investigate its effect on the reward and the population entropy, see Table 1 in Appendix F. The parameter $\alpha$ controls the balance between reward and entropy. The complete collection of the reward and population entropy learning curves with different $\alpha$ is shown in Appendix F.

**Question 3.** *What does a maximum entropy population look like?*

To have an intuition of what a maximum entropy population looks like, we show the populations in the supplementary video from 0:01 to 0:21. In this video clip, we present the population trained without the entropy reward in the first row and with the entropy reward in the second row. From the first row, we can see that the blue agent and the green agent move synchronized most of the time. The opaque routines among agents in the first population are similar. In the second row of the video clip, we can see that the behaviors of the maximum entropy population are more diverse. First, the movements of the blue agent and the green agent are less synchronized, especially in the fifth animation of the second row. Secondly, the routines are less predictable. In general, we observe more diverse behaviors and randomness of the policies in the maximum entropy population.

**Question 4.** *How does MEP perform compared to baseline methods?*

We pair each agent, including SP, PBT, and MEP, with the proxy human model $H_{Proxy}$, and evaluate the team performance. We test the performance in each of the layouts, shown in Figure 2. Good coordination between teammates is essential to achieve high scores in the collaborative game – Overcooked. Following the evaluation protocol proposed by Carroll et al. (2019), we use the cumulative rewards over a horizon of 400 timesteps as the proxy for coordination ability. For all tests, we report the average reward per episode and the standard deviation across 5 different random seeds. Figure 5 shows the quantitative results among different methods and the ablation tests. From Figure 5a, we can see that MEP outperforms both SP and PBT in all environments. Additionally, we show the ablation test in Figure 5b. From the ablation test, we can see that both the population entropy reward the prioritized sampling are necessary components for achieving the best performance.

**Question 5.** *Why does MEP perform better?*

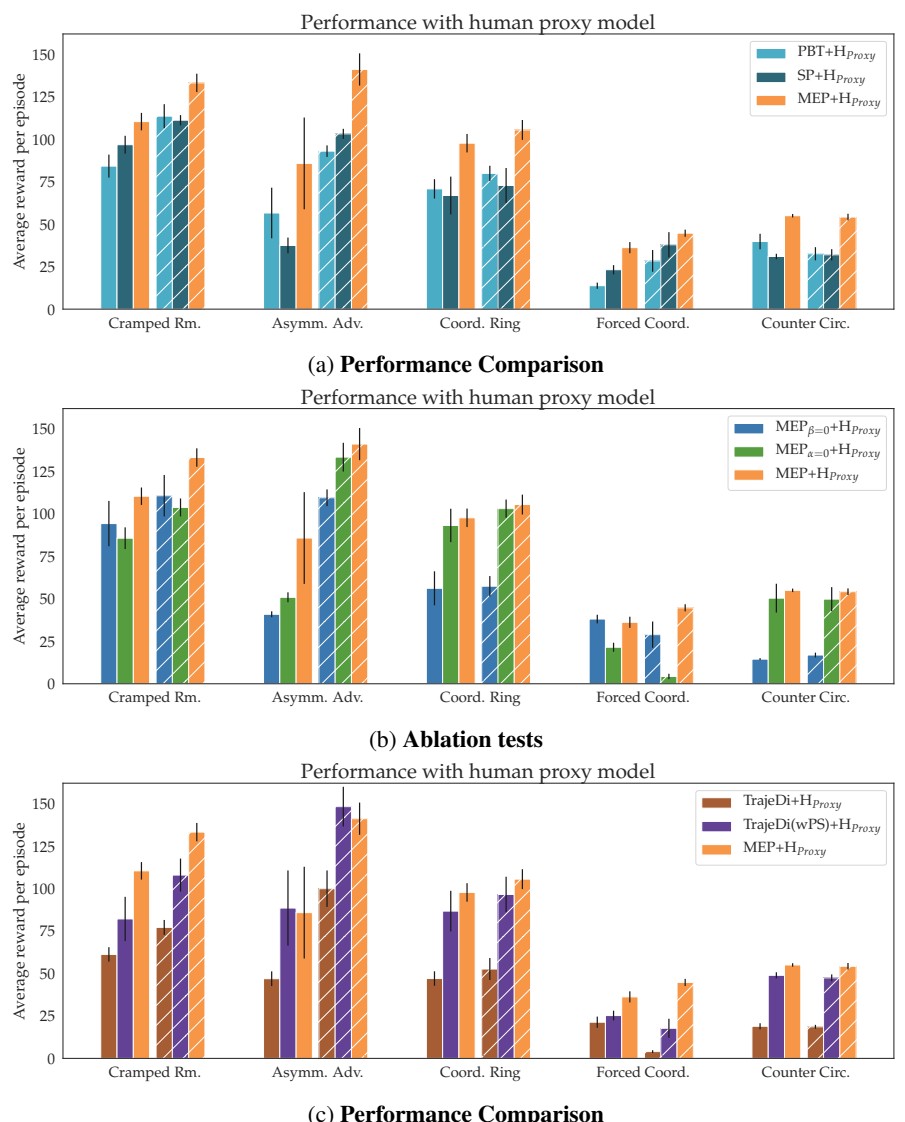

Figure 5: **Performance comparison and ablation test**: Average episode rewards over 400 timestep (1 min) trajectories for different methods, with standard error over 5 different random seeds, paired with the proxy human $H_{Proxy}$. The hashed bars with the slash (/) show results with the starting position of the agents switched. Figure (a) shows the performance comparison among MEP and baselines including SP and PBT. Figure (b) shows the ablation tests, where we use $MEP_{\alpha=0}$ and $MEP_{\beta=0}$ to denote the MEP model without the population entropy reward and without the prioritized sampling mechanism, respectively. Figure (c) shows the performance comparison with TrajeDi.

We take a closer look at some examples. The maximum entropy population contains a diverse set of policies, shown in the supplementary video from 0:01 to 0:21. In the video, from 0:24 to 0:44, we show three human-AI gameplay demonstrations of SP, PBT, and MEP, respectively. From the video, we can see that the SP-trained agent and the PBT-trained agent are prone to get stuck during the process. However, the MEP-trained agent rarely gets stuck and mostly coordinates smoothly with its human partner. The reason is that the SP-trained agent and the PBT-trained agent are more likely to overfit their training partners' policies, which are usually different from humans' strategies. However, MEP uses a high entropy population, which contains a more diverse set of policies. When the AI agent is trained using the high entropy population, it sees a wide range of behaviors and learns to adapt to these policies. This helps to make the AI agent more robust during deployment.

**Question 6.** *How does MEP perform compared to TrajeDi?*

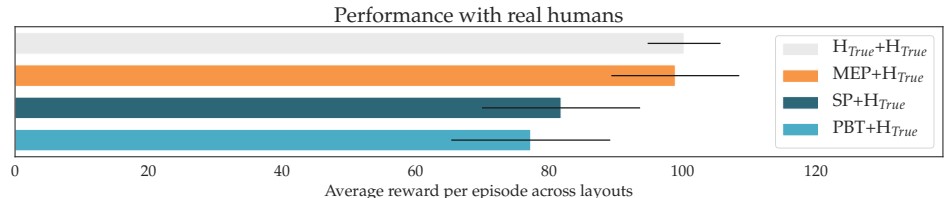

Average reward per episode in each layout

|  | Cramped Rm. | Asymm. Adv. | Coord. Ring | Forced Coord. | Counter Circ. |
|---|---|---|---|---|---|
| SP+H$_{True}$ | 094.1 | 133.2 | 084.1 | 046.8 | 050.5 |
| PBT+H$_{True}$ | 101.6 | 124.9 | 098.4 | 027.6 | 033.5 |
| MEP+H$_{True}$ | **124.1** | **154.7** | **099.5** | **056.8** | **059.1** |

Figure 6: **Performance with real humans**

There is a recent related work on zero-shot coordination with a diversified population, which is called TrajeDi (Lupu et al., 2021). To the best of our knowledge, TrajeDi is the most related work. TrajeDi utilizes a trajectory-based diversity objective to obtain a diversified population, whereas MEP formulates a novel action level diversity objective in the multi-agent population setting. To compare TrajeDi and MEP, we show the experiment results in Figure 5c. In the figure, we use TrajeDi to denote the original TrajeDi method and use TrajeDi(wPS) to denote the TrajeDi method enhanced with the proposed Prioritized Sampling (PS) mechanism. From the figure, we can see that in all settings, MEP significantly outperforms TrajeDi. In 80% settings, MEP performs superiorly in comparison to TrajeDi(wPS). Overall, MEP shows better performance compared to TrajeDi(wPS).

**Question 7.** *How does MEP perform with real human players?*

We test the MEP-trained AI agent and measured the average episode reward when the agent was paired with a real human player. For this human-AI coordination test, we recruited 40 users (24 male, 14 female, 2 other, ages 22-71) on Amazon Mechanical Turk (AMT) and followed the same evaluation procedure proposed by Carroll et al. (2019). We reuse the testing results of SP and PBT from human-AI evaluation on AMT carried out by Carroll et al. (2019). These testing results are compatible because the evaluation procedure is the same and uses a between-subjects design, meaning each user was only paired with a single AI agent. The results are presented in Figure 6. The chart in Figure 6 shows that on average across all five layouts, MEP significantly outperforms SP and PBT and its performance is on par with the Human-Human coordination performance. For more detailed results, we take a closer look at the table in Figure 6. This table shows the performance of each method in each layout. From this table, We can see that MEP achieves the best performance in all 5 layouts in comparison with SP and PBT. Now, we describe some representative cases below.

**Question 8.** *What does AI do when paired with real human players?*

Here, we show and analyze some qualitative behaviors that we observed during the real human-AI coordination experiments, which are shown in the supplementary video from 0:22 to 2:27. From 0:24 to 0:44, we observe that in the Forced Coordination layout, the MEP-trained agent is more robust and less gets stuck during coordination, in comparison to SP and PBT. Next, from 0:44 to 1:09, in the Asymmetric Advantage layout, the SP-trained and the PBT-trained agents only learned to put the onion into the pot. They didn't learn to deliver the onion soup. While, the MEP-trained agent not only learned to put the onion into the pot but also learned to deliver the onion soup when its human partner is busy. Similarly, from 1:09 to 1:29, in the Cramped Room layout, the SP-trained and PBT-trained agents only learned to use the plate to take the soup, whereas the MEP-trained agent additionally learned to carry the onion to the pot. Interestingly, from 1:29 to 1:56, in the Coordination Ring layout, the SP-trained and PBT-trained agent only learned to deliver the onion soup in one direction, while the MEP-trained agent learned to deliver the soup both clockwise and counterclockwise, depending on where its human partner stands. Last but not least, from 2:01 to 2:26, in the Counter Circuit layout, the SP-trained and PBT-trained agent learned only pass the onion over the "counter". However, the MEP-trained agent also learned to take the plate and deliver the soup. From all these observations, we observe that the SP-trained and the PBT-trained agents tend to overfit to their optimal opaque policies, whereas MEP-trained agent is more robust.

## 5 RELATED WORK

Recent works (Lerer & Peysakhovich, 2018; Tucker et al., 2020; Carroll et al., 2019; Knott et al., 2021) tackle the collaboration problem using some behavioral data from the partner to select the equilibrium of the existing agents (Lerer & Peysakhovich, 2018; Tucker et al., 2020) or build and incorporate a human model into the training process (Carroll et al., 2019; Knott et al., 2021). However, collecting a large amount of human data in real life is expensive and time-consuming. We consider the zero-shot setting, where no behavioral data from the human partner is available during training (Hu et al., 2020). From a Bayesian perspective, when we don't know what the human policies look like, we want to train the AI agent to be robust and be capable of collaborating with a diverse set of policies (Murphy, 2012). There is a growing amount of works on diversity in maximum entropy reinforcement learning (Ziebart et al., 2008; Ziebart, 2010; Fox et al., 2015; Haarnoja et al., 2017; 2018b), many of which leverage it as a means of encouraging exploration (Schulman et al., 2017a; Haarnoja et al., 2018b) or discovering skills (Eysenbach et al., 2019; Zhao et al., 2021). However, how to train a diversified population through entropy maximization is still subjective to research. In Multi-agent Reinforcement Learning (MARL), a group of agents is trained to achieve a common goal by Centralized Training and Decentralized Execution (CTDE) (Lowe et al., 2017; Foerster et al., 2018). Taking inspiration from CTDE, we propose to train a population of agents to maximize a centralized surrogate objective – population entropy, to encourage diversity in the population. Subsequently, we train the AI agent with the maximum entropy population and dynamically sample the partner agent based on the learning progress, which shares similarities with Prioritized Fictitious Self-Play (PFSP) (Vinyals et al., 2019b). PFSP is designed exclusively for zero-sum competitive games, whereas we are concerned with cooperative games and derive the relationship between prioritized sampling and cooperation performance lower bound, see Appendix C. With prioritized sampling, we make the AI agent learn a policy that is generally suitable for all the strategies presented in the population.

The idea of MEP shares a common intuition with domain randomization, where some features of the environment are changed randomly during training to make the policy robust to that feature (Tobin et al., 2017; Yu et al., 2017; Peng et al., 2018; Tan et al., 2018; Akkaya et al., 2019; Tang et al., 2020). MEP can be seen as a domain randomization technique, where the randomization is conducted over a set of partners' policies. A recent related work – TrajeDi (Lupu et al., 2021) has a similar motivation and formulates a trajectory-based diversity objective. To the best of our knowledge, TrajeDi is the most related work to MEP. In comparison to TrajeDi, we derive the population entropy objective as an action level diversity objective, which is suitable for the multi-agent PBT setting. In the experiments, MEP shows superior performance compared to TrajeDi empirically. There are also other population diversity-based methods, such as Diversity via Determinants (DvD) (Parker-Holder et al., 2020) and Diversity-Inducing Policy Gradient (DIPG) (Masood & Doshi-Velez, 2019), which are formulated for the single-agent setting, whereas MEP is designed for the multi-agent cooperative setting. In games with non-transitive dynamics where strategic cycles exist, e.g., Rock-Paper-Scissors, Policy-Space Response Oracle (PSRO)-based methods (Balduzzi et al., 2019; Perez-Nieves et al., 2021; Liu et al., 2021) provide solutions to learn diverse behaviors. Our method is complementary to these previous works and could be combined with them. MEP bridges maximum entropy RL and PBT, which is generally applicable for many human-AI coordination tasks.

## 6 CONCLUSION

This paper introduces Maximum Entropy Population-based training (MEP), a deep reinforcement learning method for robust human-AI coordination. The derived population entropy theoretical objective encourages learning a diverse set of policies. Subsequently, with the learning progress-based prioritized sampling technique, MEP helps the AI agent to be robust to different human strategies. In the simulated environments, we show that the developed approach achieves the best overall performance in comparison to state-of-the-art methods. Furthermore, in the real world evaluation with human players, MEP still demonstrates superior performance, which is comparable with human-human coordination performance. In addition, the qualitative examples show that MEP-trained policies are relatively flexible and robust to various human strategies.

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

## APPENDIX

## A   POPULATION ENTROPY LOWER BOUND

**Theorem 2.** *Let the population diversity be defined as:*

$$PD(\{\pi^{(1)}, \pi^{(2)}, ..., \pi^{(n)}\}, s_t) \coloneqq \sum_{i=1}^{n} \sum_{j \neq i}^{n} CE(\pi^{(i)}(\cdot|s_t), \pi^{(j)}(\cdot|s_t)) + \sum_{i=1}^{n} \mathcal{H}(\pi^{(i)}(\cdot|s_t)). \quad (11)$$

*Let the population entropy be defined as:*

$$PE(\{\pi^{(1)}, \pi^{(2)}, ..., \pi^{(n)}\}, s_t) \coloneqq \mathcal{H}(\bar{\pi}(\cdot|s_t)), \text{ where } \bar{\pi}(a_t|s_t) \coloneqq \frac{1}{n} \sum_{i=1}^{n} \pi^{(i)}(a_t|s_t). \quad (12)$$

*Then, we have*

$$PD(\{\pi^{(1)}, \pi^{(2)}, ..., \pi^{(n)}\}, s_t) \geq n^2 PE(\{\pi^{(1)}, \pi^{(2)}, ..., \pi^{(n)}\}, s_t), \quad (13)$$

*where $n$ is the population size.*

*Proof.* Cross entropy is given by:

$$\mathrm{CE}(\pi^{(i)}(\cdot|s_t), \pi^{(j)}(\cdot|s_t)) = -\sum_{a \in \mathcal{A}} \pi^{(i)}(a_t|s_t) \log \pi^{(j)}(a_t|s_t) \quad (14)$$

Entropy is given by:

$$\mathcal{H}(\pi^{(i)}(\cdot|s_t)) = -\sum_{a \in \mathcal{A}} \pi^{(i)}(a_t|s_t) \log \pi^{(i)}(a_t|s_t). \quad (15)$$

We can derive the follows:

$$\text{PD}(\{\pi^{(1)}, \pi^{(2)}, ..., \pi^{(n)}\}, s_t) = \sum_{i=1}^{n} \sum_{j \neq i} \text{CE}(\pi^{(i)}(\cdot | s_t), \pi^{(j)}(\cdot | s_t)) + \sum_{i}^{n} \mathcal{H}(\pi^{(i)}(\cdot | s_t)) \quad (16)$$

$$= n^2 \sum_{a \in \mathcal{A}} \frac{1}{n^2} \left( \sum_{i,j} -\pi^{(i)}(a_t | s_t) \log \pi^{(j)}(a_t | s_t) \right) \quad (17)$$

$$= n^2 \sum_{a \in \mathcal{A}} \sum_{i} -\frac{1}{n} \pi^{(i)}(a_t | s_t) \sum_{j} \frac{1}{n} \log \pi^{(j)}(a_t | s_t) \quad (18)$$

$$\geq n^2 \sum_{a \in \mathcal{A}} \sum_{i} -\frac{1}{n} \pi^{(i)}(a_t | s_t) \log \sum_{j} \frac{1}{n} \pi^{(j)}(a_t | s_t) \quad (19)$$

$$= n^2 \sum_{a \in \mathcal{A}} -\bar{\pi}(a_t | s_t) \log \bar{\pi}(a_t | s_t) \quad (20)$$

$$= n^2 \mathcal{H}(\bar{\pi}(\cdot | s_t)) \quad (21)$$

$$= n^2 \text{PE}(\{\pi^{(1)}, \pi^{(2)}, ..., \pi^{(n)}\}, s_t) \quad (22)$$

From Equation (18) to Equation (19), we use Jensen's inequality (Murphy, 2012). □

## B  POPULATION ENTROPY AND JENSEN-SHANNON DIVERGENCE

**Lemma 3.** *The population entropy has the following relation with Jensen-Shannon Divergence (JSD):*

$$\mathcal{H}(\bar{\pi}(\cdot | s_t)) = JSD(\pi^{(1)}(\cdot | s_t), ..., \pi^{(n)}(\cdot | s_t)) + \frac{1}{n} \sum_{i=1}^{n} \mathcal{H}(\pi^{(i)}(\cdot | s_t)), \quad (23)$$

*where JSD is given by*

$$JSD(\pi^{(1)}(\cdot | s_t), ..., \pi^{(n)}(\cdot | s_t)) = \frac{1}{n} \sum_{i=1}^{n} \sum_{a \in \mathcal{A}} \pi^{(i)}(a_t | s_t) \log \frac{\pi^{(i)}(a_t | s_t)}{\bar{\pi}(a_t | s_t)}. \quad (24)$$

*Proof.*

$$\mathcal{H}(\bar{\pi}(\cdot | s_t)) = \sum_{a_t \in \mathcal{A}} -\bar{\pi}(a_t | s_t) \log \bar{\pi}(a_t | s_t) \quad (25)$$

$$= \sum_{i=1}^{n} \sum_{a_t \in \mathcal{A}} -\frac{1}{n} \pi_i(a_t | s_t) \log \bar{\pi}(a_t | s_t) \quad (26)$$

$$= \sum_{i=1}^{n} \sum_{a_t \in \mathcal{A}} \frac{1}{n} \pi_i(a_t | s_t) \left( \log \frac{\pi_i(a_t | s_t)}{\bar{\pi}(a_t | s_t)} - \log \pi_i(a_t | s_t) \right) \quad (27)$$

$$= JSD(\pi^{(1)}(\cdot | s_t), ..., \pi^{(n)}(\cdot | s_t)) + \frac{1}{n} \sum_{i=1}^{n} \mathcal{H}(\pi^{(i)}(\cdot | s_t)) \quad (28)$$

□

From the derivation, we can see that maximizing the population entropy is equivalent to maximizing the JSD of all agent's policies in the population and the entropy of each agent's policy. Since the JSD measures similarity, the first term of Equation (28) encourages the policies to be different from each other. The second term of Equation (28) encourages each policy to explore.

## C  PRIORITIZED SAMPLING AND PERFORMANCE LOWER BOUND

Here, we use $\pi^{(A)}$ to denote the AI policy and use $\theta$ to represent the parameter of $\pi^{(A)}$. At the training step $t$, we use $\theta_t$ to denote the current parameter. $\{\pi^{(1)}, ..., \pi^{(n)}\}$ is the population used to train $\pi^{(A)}$. We want to find the optimal parameter $\theta^*$ for the AI policy $\pi^{(A)}$, so that the AI agent could cooperate well with any agent in the population.

At each training step, we first sample $\pi^{(i)}$ from the population, then let $\pi^{(i)}$ to cooperate with $\pi^{(A)}$. Then, we use the sampled trajectory $\tau$ to train $\pi^{(A)}$. $J(\pi^{(A)}, \pi^{(i)})$ is the excepted sum rewards achieved by $\pi^{(A)}$ and $\pi^{(i)}$ together. With prioritized sampling introduced in Section 3.4, we assign higher priority to the agent that is harder to collaborate with. To be more specific, let

$$i = \arg\min_i J(\pi_{\theta_t}^{(A)}, \pi^{(i)}), \tag{29}$$

then we sample $\pi^{(i)}$ to cooperate with $\pi_{\theta_t}^{(A)}$. During training, $\theta$ is updated using the gradient ascent method with non-increasing learning rate $\alpha_t$. At each training step $t$, there exists a subset of $\{1, ..., n\}$ denoted by $K_{\theta_t} = \{i_k | k = 1, ..., l, l \leq n\}$, such that

$$J(\pi_{\theta_t}^{(A)}, \pi^{(i)}) > J(\pi_{\theta_t}^{(A)}, \pi^{(i_k)}) = C_t, \quad \text{if } i \notin K_{\theta_t}, i_k \in K_{\theta_t}. \tag{30}$$

Since prioritized sampling is used, one of $i_{k'}$ ($k' \in 1, ..., l$) could be sampled. Then the gradient of the current step $t$ is

$$\nabla_{\theta_t} J(\pi_{\theta_t}^{(A)}, \pi^{(i_{k'})}) \tag{31}$$

The parameter is updated as following:

$$\theta_{t+1} = \theta_t + \alpha_t \nabla \theta_t, \tag{32}$$

where $\alpha_t$ is the learning rate at the training time step $t$.

Assume that $\{J(\pi_{\theta_t}^{(A)}, \pi^{(i)}) | i = 1, ..., n\}$ are smooth towards $\theta_t$, then

$$g(\theta) = \min_{i \in \{1,...,n\}} J(\pi_\theta^{(A)}, \pi^{(i)}) \tag{33}$$

is a piece-wise smooth function and $\nabla_{\theta_t} J(\pi_{\theta_t}^{(A)}, \pi^{(i_{k'})})$ is equal to the gradient of $g(\theta)$ almost everywhere. Next we prove that using $\nabla_{\theta_t} J(\pi_{\theta_t}^{(A)}, \pi^{(i_{k'})})$, it could also converge to a local maximum of $g(\theta)$.

**Lemma 4.** *$\pi^{(A)}$ with parameter $\theta$ is trained with the population $\{\pi^{(1)}, ..., \pi^{(n)}\}$. We use the learning progress-based prioritized sampling to sample the agent from the population for training. Assume that $J(\pi_\theta^{(A)}, \pi^{(i)})$ is smooth towards the parameter vector $\theta$ and has an L-Lipschitz gradient for all $i$. $\theta$ is optimized using the gradient ascent with a sufficiently small constant step size. If $J(\pi_\theta^{(A)}, \pi^{(i)})$ converges and doesn't go to infinity, it would converge to a local maximum of $g(\theta)$. That is $\theta$ converges to a neighborhood $V_{\hat{\theta}}$ of $\hat{\theta}$, where $\hat{\theta}$ is define as*

$$\hat{\theta} = \arg\max_{\theta \in V_{\hat{\theta}}} \min_{i \in \{1,...,n\}} J(\pi_\theta^{(A)}, \pi^{(i)}). \tag{34}$$

*Proof.* $\theta_t$ denotes the parameter of $\pi^{(A)}$ at the training step $t$. We define the index set of $i$, that $J(\pi_{\theta_t}^{(A)}, \pi^{(i)})$ equal to $\min_{i \in \{1,...,n\}} J(\pi^{(A)}, \pi^{(i)})$:

$$K_{\theta_t} = \{i_k | k = 1, ..., l, l \leq n\}, \tag{35}$$

where $i_k$ satisfies

$$J(\pi_{\theta_t}^{(A)}, \pi^{(i)}) > J(\pi_{\theta_t}^{(A)}, \pi^{(i_k)}) = C_t, \quad \text{if } i \notin K_{\theta_t}, i_k \in K_{\theta_t}. \tag{36}$$

$i_{k'}$ is sampled from $K_{\theta_t}$ and current gradient is $\nabla_{\theta_t} J(\pi_{\theta_t}^{(A)}, \pi^{(i_{k'})})$.

If $i_{k'} \in K_{\theta_{t+l}}$ for all $l > 0$, the optimization process could be regarded as a non-convex optimization problem by gradient ascent, then $\theta_t$ converges to a local maximum almost surely by the assumption $J(\pi_{\theta_t}^{(A)}, \pi^{(i)})$ has an L-Lipschitz gradient (Lee et al., 2016).

If $\bigcap_{l>0} K_{\theta_{t+l}} = \emptyset$, then first we prove that the sequence $\{\theta_t\}$ can't converge to a saddle point. If $\hat{\theta}$ is a saddle point, $\nabla_\theta J(\pi_{\hat{\theta}}^{(A)}, \pi^{(i_{k'})}) = 0$ for all $i_{k'} \in K_{\hat{\theta}}$, which means $J(\pi_{\theta_t}^{(A)}, \pi^{(i_{k'})})$ are identical in a neighborhood of $\hat{\theta}$. This contradicts the local minimum convergence of $\theta_t$ (Lee et al., 2016).

Assume the convergent point $\hat{\theta}$ has a non-zero gradient $\nabla_\theta J(\pi_{\theta_t}^{(A)}, \pi^{(i_{k'})})$, since we use a sufficiently small constant learning rate $\alpha$, if $J(\pi_{\hat{\theta}+\Delta\theta}^{(A)}, \pi^{(i_{k'})}) > J(\pi_{\hat{\theta}}^{(A)}, \pi^{(i_{k'})})$, this would contradict the convergence of $\theta$, and if $J(\pi_{\hat{\theta}+\Delta\theta}^{(A)}, \pi^{(i_{k'})}) \leq J(\pi_{\hat{\theta}}^{(A)}, \pi^{(i_{k'})})$, which means $\hat{\theta}$ is the local maximum.

$\square$

From Lemma 4, we can see that with prioritized sampling, we could improve the lower bound of the cooperation performance between $\pi^{(A)}$ and the population $\{\pi^{(1)}, ..., \pi^{(n)}\}$. In comparison, uniform sampling does not provide any guarantee on the worst case. We call $\theta'$ the optimal solution of mean sampling:

$$\theta' = \underset{\theta}{\arg\max} \sum_{i \in \{1,...,n\}} J(\pi_\theta^{(A)}, \pi^{(i)}). \tag{37}$$

Then, the worst cooperation between $\pi_{\theta'}$ and the population must be no greater than the cooperation between $\pi_{\hat{\theta}}$ and the population. That is:

$$\min_{i \in \{1,...,n\}} J(\pi_{\hat{\theta}}^{(A)}, \pi^{(i)}) \geq \min_{i \in \{1,...,n\}} J(\pi_{\theta'}^{(A)}, \pi^{(i)}). \tag{38}$$

## D    RELATION TO HUMAN-AI COORDINATION PERFORMANCE

To illustrate that we could improve the lower bound of human-AI coordination performance, here we introduce the connection between $\epsilon$-close (Ko, 2006) and return, i.e., expected sum rewards.

**Definition 1.** *We define that $\pi^{(1)}$ is $\epsilon$-close to $\pi^{(2)}$ at the state $s_t$ if*

$$\left| \frac{\pi^{(1)}(a_t|s_t)}{\pi^{(2)}(a_t|s_t)} - 1 \right| < \epsilon \tag{39}$$

*for all $a_t \in \mathcal{A}$. If this is satisfied at every $s_t \in \mathcal{S}$, we call $\pi^{(1)}$ is $\epsilon$-close to $\pi^{(2)}$.*

**Lemma 5.** *If an MDP has $T$ time steps and $\pi^{(1)}$ is $\epsilon$-close to $\pi^{(2)}$, then for all $\pi^{(A)}$, we have*

$$(1-\epsilon)^T J(\pi^{(2)}, \pi^{(A)}) < J(\pi^{(1)}, \pi^{(A)}) < (1+\epsilon)^T J(\pi^{(2)}, \pi^{(A)}), \tag{40}$$

*where $J(\pi^{(i)}, \pi^{(A)})$ denotes the expected sum reward achieved by $\pi^{(i)}$ and $\pi^{(A)}$ collaborating with each other.*

*Proof.* For all trajectory $\tau$, we have

$$p_{\pi^{(1)}, \pi^{(A)}}(\tau) = p(s_0)\Pi_{t=0}^{T-1} \pi^{(A)}(a_t|s_t)\pi^{(1)}(a_t|s_t)p(s_{t+1}|s_t, a_t^{(A)}, a_t^{(1)}). \tag{41}$$

Since

$$(1-\epsilon)\pi^{(2)}(a_t|s_t) < \pi^{(1)}(a_t|s_t) < (1+\epsilon)\pi^{(2)}(a_t|s_t), \tag{42}$$

we have

$$(1-\epsilon)^T p_{\pi^{(2)}, \pi^{(A)}}(\tau) < p_{\pi^{(1)}, \pi^{(A)}}(\tau) < (1+\epsilon)^T p_{\pi^{(2)}, \pi^{(A)}}(\tau). \tag{43}$$

And because

$$J(\pi^{(i)}, \pi^{(A)}) = \sum_\tau p_{\pi^{(1)}, \pi^{(A)}}(\tau)r(\tau), \tag{44}$$

where

$$r(\tau) = r(s_0, a_0, ..., s_{T-1}) = \sum_t r(s_t, a_t) \text{ and } a_t = (a^{(i)}, a^{(A)}). \tag{45}$$

Therefore, we have

$$(1 - \epsilon)^T J(\pi^{(2)}, \pi^{(A)}) < J(\pi^{(1)}, \pi^{(A)}) < (1 + \epsilon)^T J(\pi^{(2)}, \pi^{(A)}). \tag{46}$$

$\square$

We use $\pi^{(H)}$ to denote the human player's policy. From Lemma 5, we can see that if $\pi^{(H)}$ is similar to any of the policy $\pi^{(i)}$ in the population $\{\pi^{(1)}, ..., \pi^{(n)}\}$ in a certain degree, measured by $\epsilon$-close, then its cooperation performance with the AI policy $\pi^{(A)}$, which is trained with the population, would not deteriorate too much. Furthermore, we derive the following corollary.

**Corollary 1.** *We call the infimum of expected sum rewards of $\pi^{(A)}$ cooperating with the population $\{\pi^{(1)}, ..., \pi^{(n)}\}$ as:*

$$\min_{i \in \{1,...,n\}} J(\pi^{(A)}, \pi^{(i)}) = C. \tag{47}$$

*If $\pi^{(H)}$ is $\epsilon$-close to the policy $\pi^{(i)}$ in the population, then we have*

$$J(\pi^{(A)}, \pi^{(H)}) > C(1 - \epsilon)^T, \tag{48}$$

*where $T$ is the total steps in the trajectory.*

*Proof.* If $\pi^{(H)}$ is $\epsilon$-close to $\pi^{(i)}$, based on the property of $\epsilon$-close, see Lemma 5, we have

$$J(\pi^{(A)}, \pi^{(H)}) > (1 - \epsilon)^T J(\pi^{(A)}, \pi^{(i)}). \tag{49}$$

Additionally, since

$$J(\pi^{(A)}, \pi^{(i)}) > C, \tag{50}$$

we have

$$J(\pi^{(A)}, \pi^{(H)}) > C(1 - \epsilon)^T. \tag{51}$$

$\square$

Since prioritized sampling optimizes the lower bound of expected sum rewards of the AI agent cooperating with the population, see Lemma 4, and with Corollary 1, we could say that it also optimizes the lower bound of expected sum rewards of the AI agent cooperating with the Human player, when the population is diverse and representative enough so that it is close to cover human behaviors.

## E    EXPERIMENT DETAILS

We ran all the methods in each environment with 5 different random seeds and report the average episode reward and the standard deviation. The experiments of the maximum entropy population-based training use the following hyper-parameters:

- The learning rate is 8e-4.
- The reward shaping horizon is 5e6.
- The environment steps per agent is 1.1e7.
- The number of mini-batches is 10.
- The mini-batch size is 2000.
- PPO iteration timesteps are 40000. The PPO iteration timesteps refer to the length in environment timesteps for each agent pairing training.

- The population size is 5 for training the maximum entropy population. We use the beginner model, the middle model, and the best model of each agent in the population to form the final population for training the AI agent.
- The weight $\alpha$ for the population entropy reward is 0.01 in general. For the Forced Coordination layout, we use 0.04.
- The number of parallel environments used for simulating rollouts is 50.
- The discounting factor $\gamma$ is 0.99.
- The max gradient norm is 0.1.
- The PPO clipping factor is 0.05.
- The number of hidden layers is 3.
- The size of hidden layers is 64.
- The number of convolution layers is 3.
- The number of filters is 25.
- The value function coefficient 0.1.
- The $\beta$ for prioritized sampling is 3.

## F EXPERIMENTAL RESULTS

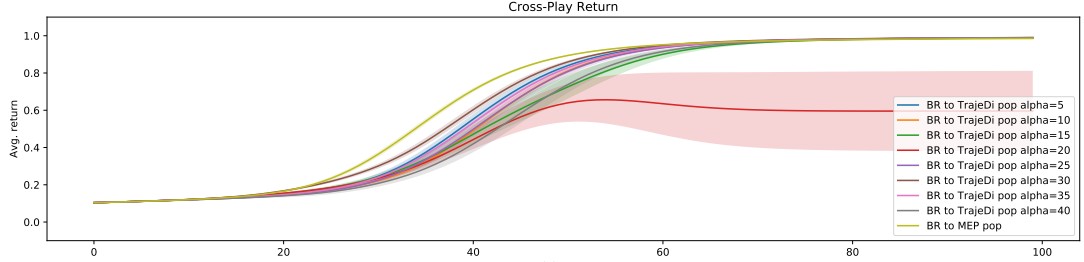

Figure 7: **Performance comparison**: We did an extensive hyper-parameter search for TrajeDi. MEP converges faster than TrajeDi under all the parameters.

Table 1: **Best self-play reward and its corresponding population entropy with different** $\alpha$: In this table, $\alpha$ denotes the weight of the population entropy reward in Equation (8).

| $\alpha$ | Cramped Rm. | | Asymm. Adv. | | Coord. Ring | | Forced Coord. | | Counter Circ. | |
|---|---|---|---|---|---|---|---|---|---|---|
| | Rew. | Ent. | Rew. | Ent. | Rew. | Ent. | Rew. | Ent. | Rew. | Ent. |
| 0.000 | 196.8 | 0.971 | 164.1 | 1.120 | 183.4 | 0.878 | 149.4 | 0.970 | 129.0 | 0.988 |
| 0.001 | 187.6 | 1.031 | 160.9 | 1.051 | 183.7 | 0.907 | 152.0 | 0.858 | 118.0 | 1.152 |
| 0.005 | 189.8 | 0.949 | 153.0 | 1.075 | 164.1 | 0.901 | 163.7 | 0.889 | 119.2 | 1.038 |
| 0.010 | 183.7 | 1.057 | 151.8 | 1.139 | 167.8 | 0.840 | 151.2 | 1.079 | 136.5 | 1.151 |
| 0.020 | 174.2 | 1.029 | 149.7 | 1.074 | 157.8 | 0.947 | 137.8 | 1.093 | 121.7 | 1.171 |
| 0.030 | 154.0 | 1.134 | 138.7 | 1.203 | 153.6 | 1.028 | 133.6 | 0.957 | 0.130 | 1.715 |
| 0.040 | 137.0 | 1.194 | 135.0 | 1.353 | 125.7 | 1.122 | 081.0 | 1.460 | 0.000 | 1.791 |
| 0.050 | 137.1 | 1.127 | 118.4 | 1.364 | 129.6 | 0.996 | 024.5 | 1.703 | 0.000 | 1.791 |

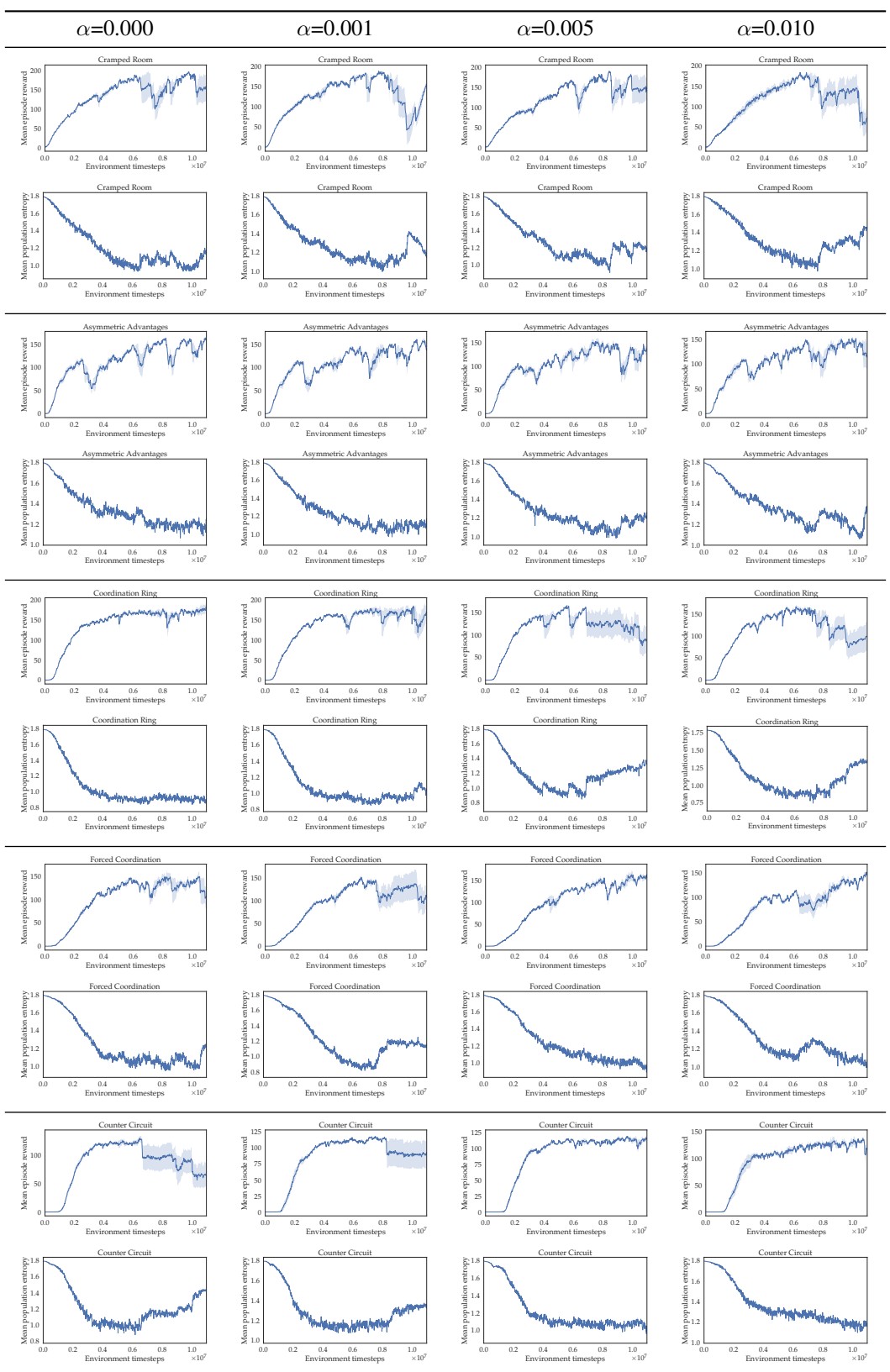

Table 2: **Mean episode reward and population entropy with different** $\alpha$ **in all five layouts:** Each column corresponds to a different value of $\alpha$ in the set of $[0.000, 0.001, 0.005, 0.010]$. There are five row sections, which correspond to the five layouts. Each row section contains two rows, which are the plots of the mean episode reward and the mean population entropy of the layout, respectively.

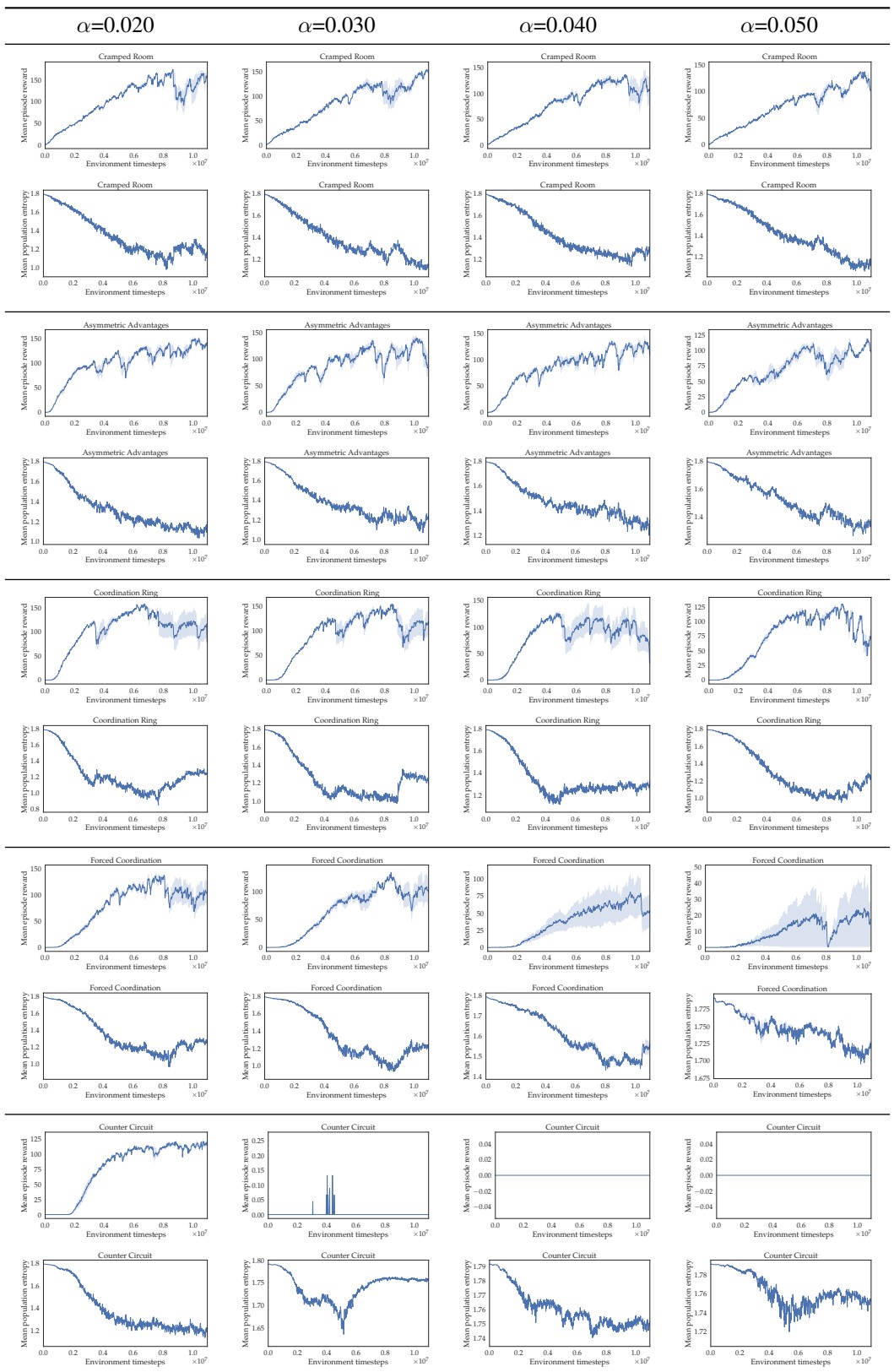

Table 3: **Mean episode reward and population entropy with different $\alpha$ in all five layouts:** Each column corresponds to a different value of $\alpha$ in the set of $[0.020, 0.030, 0.040, 0.050]$. There are five row sections, which correspond to the five layouts. Each row section contains two rows, which are the plots of the mean episode reward and the mean population entropy of the layout, respectively.

