# OpenReview forum: "Maximum Entropy Population Based Training for Zero-Shot Human-AI Coordination"
_ICLR.cc/2022/Conference — ICLR 2022 Submitted_

### Official Review · Reviewer_mavT · 2021-11-01

**Correctness:** 3
**Technical Novelty And Significance:** 2
**Empirical Novelty And Significance:** 2
**Recommendation:** 3
**Confidence:** 5

**Main Review:**

STRENGTHS

The high-level motivation of injecting diversity into partner populations is a sound and important one for human-AI coordination. The paper goes beyond just generalization to held-out agents and evaluates with real humans, which I think is crucial for this line of work. The authors include in their supplementary data videos of gameplay, and additionally discuss qualitative aspects of agent behavior, which I found useful in guiding intuition.

WEAKNESSES

There are several crucial missing or misunderstood ablations and baselines.

**Misunderstood baseline - TrajeDi**: the JSD term in the authors' population entropy bonus in equation 7 is just a special case of TrajeDi with gamma=0 (see equation 5 and discussion following in the TrajeDi paper). Thus, the population entropy bonus just amounts to adding the usual (per-agent) entropy bonus to TrajeDi. Can the authors comment? In addition, I could not find the hyperparameters or tuning procedure for the TrajeDi baseline. What value of gamma was used and how was it selected? What size population was used for TrajeDi? Comparing Fig 4a and 4c, it looks like the TrajeDi agent performs significantly worse than self-play, which suggests it was poorly tuned.

Several times the authors suggest they compare to "state-of-the-art" baselines, but the two methods that to my knowledge have a claim to this are not included.

**Missing baseline #1 - PPO_BC**: while the authors include several baselines from the Carroll et al 2019 paper, they for some reason do not include the best-performing one - the best response to a human BC model, PPO_BC. The authors might rightly protest that the comparison wouldn't be exactly apples to apples, since PPO_BC gets to use human data in training. For this reason, I don't think its a prerequisite of a human data-free method to *beat* PPO_BC, but it is still an important, informative, and easy baseline to include for comparison.

**Missing baseline #2 - FCP**: more recently, Strouse at al 2021 (https://arxiv.org/abs/2110.08176) introduced fictitious co-play (FCP), which achieved state-of-the-art human-AI coordination on Overcooked without human data, even seemingly beating the human BC model best response agent above. FCP similarly involves a two-stage training process of first training a diverse set of partners, and then training a best response to them. However, FCP achieves diversity solely through seeds of self-play, and taking checkpoints from different points of training (beginning, middle, and end). Strangely, buried in appendix E, the present authors mention that they too train their best response agent with multiple checkpoints of their partner population, again from the beginning, middle, and end of training. First, this seems like an odd and important detail to bury in an appendix and not mention in the main text. Second, it is an important enough detail to deserve an ablation (which the FCP paper did). Moreover, the combination of omitting mention of FCP while simultaneously using an eerily similar trick and not mentioning it in the main text looks pretty suspicious. In any case, from what I can tell, FCP is essentially the authors' present setup but with alpha=0, beta=0, and a bigger population (N=32 vs N=5) (though the RL algorithms also differ, i.e. VMPO vs PPO). In other words, if the authors use larger partner populations, are the population entropy bonus and prioritized sampling still important?

**Missing ablation #1 - past checkpoints**: as mentioned in the paragraph above, the authors mentioned in appendix E that they train with multiple checkpoints of their partner population. That seems like an important detail to ablate.

**Missing ablation #2 - two terms in population entropy**: the authors' population entropy objective (equation 7) contains two terms - a JSD term and the usual per-agent entropy bonus. The latter is a standard addition in many RL setups. The alpha=0 ablation in Figure 4b turns off both the JSD and per-agent entropy terms together. Thus, it is unclear whether the JSD term, the per-agent entropy bonus, or the combination is important.

**Question about uniform sampling ablation**: I was surprised to see this version do so poorly. Is it just that training takes a bit longer with uniform sampling? Had this agent converged? I couldn't find learning curves for this ablation.

**Missing baselines and ablations in human experiments**: none of the important baselines or ablations were evaluated with humans. While results with the human proxy model are helpful and suggestive, I think its important to include the key baselines and ablations in the human experiments as well, i.e. TrajeDi, alpha=0, beta=0, and the other baselines and ablations mentioned above.

**Between-subjects design**: This worries me, since I'm guessing the evaluation of the MEP agent and the original Carroll et al 2019 agents took place at least two years apart, and may have been performed by different authors. Unfortunately with human evaluations, many small details matter, such as server lag, UI, and sample population, all of which are hard to control for when evaluating two years apart. Could the authors provide additional justification for how they ensure the evaluations are identical? If not, and since I would like to see other baselines and ablations evaluated with humans anyway, perhaps it would be easy to re-evaluate SP and PBT.

Lastly, a few more minor points:
1. Structurally, I found section 3.1 and Theorem 1 unnecessary. It would be simpler to just start their narrative with equation 7, which at least to me is more well motivated than equation 2 anyway.
2. The JSD term in equation 7 is also closely related to the Emergence of Individuality (EOI) reward of Jiang & Lu 2021 (https://arxiv.org/abs/2006.05842), since the JSD term can also be understood as the mutual information between agent index and action choice, conditioned on state, or I(i;a|s).
3. In the answer to Question 1, the authors state that as alpha increases from 0 to .01, the population entropy increases while the reward barely decreases. However, from what I can tell, the changes are of similar magnitude, with reward dropping about 20% and entropy increasing 20%, so this seems like a misleading claim.
4. In the answer to Question 3, the authors state that MEP outperforms SP and PBT in all environments. However, the error bars seem to overlap on Forced, so I might change this to say "performs as well or better."
5. In the intro, are cooperative games and emergent communication really "real world applications"?
6. Tylkin et al 2020 (https://econcs.seas.harvard.edu/files/econcs/files/tylkin_neurips20.pdf) is relevant for citation and discussion on training agents for human-AI coordination.
7. MAVEN from Mahajan et al 2019 (https://arxiv.org/abs/1910.07483), as well as EOI mentioned above, are missing related work on applying population diversity-based methods in the multi-agent setting.
8. TrajeDi is described as "concurrent" work, but the paper has been out for several months. "Concurrent" is I suppose of debatable definition, but I think its more appropriate for work that is in submission at the same time.
9. There are many typos and grammatical errors in the paper. They did not damage readability for me, and I do not penalize the authors for this, but the paper would benefit from a proofreading. Some examples: "from the max ent RL" -> "by max ent RL" (abstract), "assistant" -> "assist" (paragraph 1 of intro), "use" -> "using" (paragraph  4 of intro), "incentive" -> "diverse"? (paragraph 1 of sec 3.1), "unbound" -> "unbounded" (last paragraph of sec 3.1), "Compare" -> "Compared" and "Take" -> "Taking" (paragraph after equation 6), "panic" -> "panicked" (sentence before sec 3.4), etc etc.

**Summary Of The Paper:**

The authors focus on training agents for zero-shot human-AI coordination on Overcooked. They propose to first train a population of agents with a population entropy bonus, and then train another agent as the best response to the population, using a prioritized sampling approach that focuses more training on the worst performing pairs. They evaluate their agent and some baselines and ablations both with a human proxy model (i.e. an agent trained on human-human gameplay data with behavioral cloning) and also a smaller subset of their agent and baselines with real humans on Mechanical Turk. Their method performs as well or better in terms of game score than the baselines and ablations presented.

**Summary Of The Review:**

There are several crucial missing baselines and ablations and clarifications before I think the paper is ready for publication. However, pending their inclusion and a review of the new results, I am very open to raising my score to an accept.

---

> ### Author Response · Authors · 2021-11-12
> **To Reviewer mavT**
>
> Thank you for your feedback!
>
> - There is an important technical difference between MEP and TrajeDi. TrajeDi suggests optimizing the diversity surrogate loss \^{JSD} analytically, see the third paragraph of Section 4.4 of the TrajeDi paper, using the derived gradient of \^{JSD} to update the model and encourage diversity. MEP simply adds the population entropy bonus to the original reward and implements a different way to do JSD+entropy update, which is compatible with SGD and simplifies the algorithm by removing the necessity of implementing the gradient of \^{JSD} loss from TrajeDi. Compared to TrajeDi, MEP is easier to implement and shows better empirical results in the matrix game and the Overcooked environment.
> We newly added the comparison between MEP and TrajeDi on the matrix game in the updated version. We use the same evaluation protocol as proposed by Lupu et al 2021. From Figure 3 in the updated paper, we can see that MEP converges faster than TrajeDi. We did an extensive hyper-parameter search for TrajeDi, as shown in Figure 7 in Appendix F.
>
> - We use gamma as 1 in the Overcooked game environment to compare the trajectory-based diversity with population-entropy-based diversity.
> - We use the same population size 15 (5x3) for MEP and TrajeDi. The population size is 5 for training the maximum entropy population. We use the beginner model, the middle model, and the best model of each agent in the population to form the final population for training the AI agent. The final population size for training the AI agent is 15.
>
> - TrajeDi without prioritized sampling performs worse than self-play because of the diversity of the population. We also tested the TrajeDi with prioritized sampling (TrjaeDiwPS), which significantly outperforms self-play, as shown in Figure 4 (a) and (c). In comparison with TrajeDiwPS, MEP is still better in 80% of the environments. We believe that we made a fair comparison.
>
> - About PPO_BC: We did not include PPO_BC for comparison because PPO_BC uses human data for training. All the other methods, including PPO, PBT, MEP, and TrajeDI, do not use human data. We tested all the methods in a zero-shot fashion. For the performance of PPO_BC, readers can refer to the original paper written by Carroll et al. 2019. Besides, we did present the real Human-Human coordination performance in Figure 6 in the updated paper, which is more realistic than PPO_BC.
>
> - About FCP: FCP written by Strouse et al. 2021 (https://arxiv.org/abs/2110.08176) is released on arXiv on **15 Oct 2021**. However, the ICLR 2022 Submission date is **Oct 05, 2021**. After reading the FCP paper, we found out that the method MEP_{\alpha=0} in the ablation study, see Figure 5 (b), is essentially the FCP baseline, where alpha=0 means no population entropy but with prioritized sampling, i.e., FCP. Here, the population size is 5x3=15. When the population size is 15, our method outperforms FCP. We believe the reason is that the population entropy brings better diversity to the population given the same population size.
>
> - About past checkpoints: In Figure 5 (b), the result of MEP_{\alpha=0} (\alpha=0 means no population entropy reward) represents the experiment result with checkpoints from different points of training, i.e., beginning, middle, and end. In comparison with the result of MEP in the same figure, we can see that the population entropy indeed increases the performance.
>
> - About two terms in population entropy: In the current paper, we consider the population entropy, which consists of JSD and per-agent entropy, as a whole term. The effects of the JSD and the per-agent entropy terms are complementary. The JSD term increases the mutual difference among policies in the population. The per-agent entropy term increases the diversity of each individual’s policy. We  leave the study of different variants of population entropy to future work.
>
> - About uniform sampling: We train the agents with uniform sampling for the same amount of time as we train with prioritized sampling. The agents have converged.
>
> - We didn't run all the baselines and ablation in real human experiments because real human experiments are too expensive and time-consuming. We run all the experiments with the human proxy models. The results with the human-proxy models are similar to the real human experiments in most cases [Carroll et al. 2019.].
>
> - About the between-subjects design: we use the same code and evaluation protocol as the original work by Carroll et al. 2019. Moreover, we confirmed with Carroll et al. that we can run real human experiments only with the newly proposed method.
>
> - Thank you for pointing out the work Emergence of Individuality (EOI), which provides a new interpretation of the JSD term.
>
> - We fixed some of the minor points addressed by the reviewer in the updated version of the paper. Furthermore, we change the phrase “concurrent work” to “recent related work” regarding TrajeDi. We also fixed the typos and grammatical errors.

---

> > ### Comment · Reviewer_mavT · 2021-11-21
> > **Thanks for your response**
> >
> > Thanks for your response. While it is helpful, I still have remaining concerns and questions.
> >
> > 1. I still think it is insufficient to evaluate Trajedi with gamma=1 only on Overcooked. This is a key hyperparameter that should be fairly tuned, and indeed from what I can tell in the original paper, the best value on Hanabi was not even gamma=1 (but gamma=0.5). While I understand the temptation to choose the value of gamma that makes Trajedi most dissimilar to MEP and thus increases the perceived novelty of MEP, it should also be noted that there is already some novelty in evaluating on a new domain (i.e. Overcooked vs Hanabi).
> > 2. Why is the inclusion of multiple checkpoints not discussed / motivated in the main text? Also, why is not ablated? Again, this seems like an important detail to bury.
> > 3. I still maintain that it is important to include all key baselines in the human evaluation. As is, MEP is not compared to any competitive baselines. As I said originally, PPO_BC should be included, even if it is ok for MEP not to beat it, since it of course has the unfair advantage of additionally using human data. Additionally, Trajedi and key ablations should be included. The marginal expense of evaluating another agent after setting up the evaluation pipeline is modest and other works have found it feasible to evaluate more agents (e.g. 3 agents in Carroll et al 2019 and 5 agents in Strouse et al 2021).
> > 4. Can you elaborate on what you mean by: human-human scores are "more realistic than PPO_BC"?
> > 5. The authors fairly pointed out that the FCP paper came out after their submission, and so a direct experimental comparison for this draft is unfair. I also appreciated the point that alpha=0 MEP is close to being an FCP baseline. That said, the FCP paper used larger populations (32 vs 5). Speculating a bit, my guess is that if the population sizes were increased, FCP and MEP would perform similarly, since the diversity induced by the different seeds and checkpoints will be "enough." However, with small populations, it may well be that MEP performs better than FCP, since it more explicitly encourages diversity. Of course, FCP is "simpler" in that no additional intrinsic rewards are required, so this would present an interesting tradeoff (i.e. implementation complexity vs needing to train bigger populations) if true.
> > 6. I still think it would improve the scientific contribution of the paper to separately ablate the two terms in the population entropy, since one of them is shared with Trajedi (gamma=0).

---

### Official Review · Reviewer_MRtf · 2021-11-01

**Correctness:** 2
**Technical Novelty And Significance:** 2
**Empirical Novelty And Significance:** 3
**Recommendation:** 5
**Confidence:** 4

**Main Review:**

At present I do not believe this paper should be published at ICLR, for the following reason: it claims to present a novel approach yet it simply combines two previously published methods (TrajeDi and PFSP). The framing of the method is incorrect, it does not cite PFSP until related work (not in Section 3.4) and it claims TrajeDi is "concurrent" which is not true. TrajeDi was presented at AAMAS 2021 (in May) and again at ICML 2021 (in July). This to me is not concurrent. To be concrete on the differences, the TrajeDi objective is almost identical, when the $\gamma$ parameter is set to zero. Meanwhile, PFSP uses almost the same prioritization scheme, but the only difference is that it is used in competitive games vs. cooperative.

However, I believe there is material in this paper that is worthy of publication, and a simple reframing/restructuring may be sufficient for the paper to be a useful contribution. Indeed, the paper does present novel theoretical results, a novel *combination* of existing methods in a new setting (since PFSP has only been used in competitive settings until now), and interesting experiments including testing their agents with humans. This could be a good paper if it reduces claims of novelty and instead focuses on how it builds on these previous works. **If significant changes are made to the claims in the paper and presentation of the methods I would be happy to raise my score**.

More Detailed Comments:
* The ablation studies are useful. Particularly the inclusion of the prioritization scheme with TrajeDi. Which value for $\gamma$ was used for TrajeDi? I would guess it is using $\gamma=1$ which means the results show us the difference between action diversity and trajectory diversity, which is an interesting result, but likely problem specific. What happens if you use other hyperparameters for TrajeDi?
* Figure 3 has a lot of redundancy. I would move both the episode reward curves onto the same plot and also move both of the entropy curves onto the same plot, then use the colors to differentiate between the methods and a shared legend. Then have the name of the plot on the title (across the top) rather than as a y-axis label.
* Missing baseline: Off Belief Learning (Hu et al 2021) is the state-of-the-art method for ZSC in Hanabi. It would be interesting to see how it performs here. At minimum it should be cited.
* For the hyperparameter studies in the Appendix, it would be great if we can also compare different configurations on the same plots with the same axis.
* As always, it is great that code is included.
* The human experiments are interesting. However, it seems the baselines are reduced. How does TrajeDi perform here? Does the addition of PFSP impact the human coordination? It seems like there are a lot of unanswered questions. Nonetheless, as it is this is a strong result.
* There are no discussion of limitations in the paper. Honest discussion of this would make the work stronger. For example, how would it work with larger population sizes and higher dimensional problems (such as Hanabi).
* Figure 5 - my guess is the bolding just means a higher average, without considering the error bars. It is better to only bold if the error bars do not overlap.

**Summary Of The Paper:**

This paper proposes an approach for training agents that are capable of ad hoc coordination with humans. The method combines an entropy based objective with a method for prioritizing partner selection. Results on the Overcooked domain show improved performance over a series of baselines.

**Summary Of The Review:**

The paper is interesting and well written, and addresses an important problem. The method itself combines two known methods (TrajeDi and PFSP) but does not provide sufficient credit, since it claims TrajeDi is "concurrent" and it only discusses PFSP in related work. Accurately positioning the new contribution w.r.t the previous would make this a solid contribution.

---

> ### Author Response · Authors · 2021-11-12
> **To Reviewer MRtf**
>
> Thank you for your detailed feedback!
>
> - We now cite PFSP in Section 3.4 and replace the phrase “a concurrent work” with “a recent related work” regarding TrajeDi.
>
> - The motivation between our method MEP and TrajeDi are different.
> MEP is motivated by maximum entropy RL, takes inspiration from the CTDE framework in Multi-agent RL, and derives the population entropy objective. While TrajeDi is motivated by encouraging diversity in the trajectory-level instead of the state-level. Even with \gamma set to 0, there is still a difference between TrajeDi and MEP. MEP additionally encourages individual diversity.
> We newly added the comparison between MEP and TrajeDi on the matrix game in the updated version.  In the single-step collaborative matrix game, player 1 must select a row while player 2 chooses a column independently. Both agents get the reward associated with the intersection of their choices at the end of the game. We use the same evaluation protocol as proposed by Lupu et al 2021. From Figure 3 in the updated paper, we can see that MEP converges faster than TrajeDi. We did an extensive hyper-parameter search for TrajeDi, as shown in Figure 7 in Appendix F.
>
> - Yes, we used gamma as 1 for the "Overcooked" game experiments. In the matrix game, since it is a one-step environment, the gamma has to be 0.
>
> - For human experiments, we only tested with MEP because human experiments are too expensive and time-consuming. Besides, the results with the human-proxy models are similar to the human experiments in most cases.

---

> > ### Comment · Reviewer_MRtf · 2021-11-14
> > **Shifting emphasis for the review**
> >
> > Thank you for this update.
> >
> > I am increasing my score from a "reject" -> "weak reject" as the authors now appropriately reference TrajeDi and PFSP. I am no longer concerned about the issue of falsely claiming novelty.
> >
> > At present, the reason for not switching to "weak accept" is because I am not convinced there is much new here. It seems like this is just TrajeDi + entropy bonus + PFSP. I don't buy the argument that "the methods are different because the motivation is different". I think it might also be helpful to include different values of $\gamma$ for TrajeDi in the main experiments, to show the impact of the entropy bonus and PFSP without the confounder of $\gamma$.
> >
> > In the meantime, I will spend some more time reading through the paper and responses to other reviews.

---

> > > ### Author Response · Authors · 2021-11-15
> > > **To Reviwer MRtf**
> > >
> > > Thank you for reading our rebuttal and updating the review!
> > >
> > > - To your comment "At present, the reason for not switching to "weak accept" is because I am not convinced there is much new here."
> > > There is an important technical difference between MEP and TrajeDi.
> > > TrajeDi suggests optimizing the diversity surrogate loss \\^{JSD} analytically, see the third paragraph of Section 4.4 of the TrajeDi paper, using the derived gradient of \\^{JSD} to update the model and encourage diversity.
> > > MEP simply adds the population entropy bonus to the original reward and implements a different way to do JSD+entropy update, which is compatible with SGD and simplifies the algorithm by removing the necessity of implementing the gradient of \\^{JSD} loss from TrajeDi.
> > > Compared to TrajeDi, MEP is easier to implement and shows better empirical results in the matrix game and the Overcooked environment.

---

### Official Review · Reviewer_1grX · 2021-11-03

**Correctness:** 4
**Technical Novelty And Significance:** 3
**Empirical Novelty And Significance:** 2
**Recommendation:** 6
**Confidence:** 3

**Main Review:**

Strength:

1. The paper is well-written and easy to follow.
2. The idea of combing the population entropy during the training is interesting, which considers both agents' individual diversity and pairwise diversity.

Concerns:
1. The authors propose a good way to promote the policy diversity in multi-agent RL, and this is straightforward that the diversity policy would benefit the zero-shot human-ai coordination. However, I think policy diversity is a general method that can be useful in many other problems, not only limited to human-ai coordination. Therefore, I think the authors can discuss the broader impact of the paper and clarify more about why test this method on specific human-ai coordination problem.
2. The prioritized sampling + entropy population seems important to train the diverse policy, and an ablation study to investigate how each component works would be better.
3. Some important references about diversity in population-based multi-agent RL are missing, like [1,2,3]. It would be good to discuss the relationship between the proposed method and the diversity promoting solutions in the PSRO framework.

[1] Balduzzi, David, et al. "Open-ended learning in symmetric zero-sum games." International Conference on Machine Learning. PMLR, 2019.

[2] Nieves, Nicolas Perez, et al. "Modelling behavioural diversity for learning in open-ended games." arXiv preprint arXiv:2103.07927 (2021).

[3] Liu, Xiangyu, et al. "Unifying Behavioral and Response Diversity for Open-ended Learning in Zero-sum Games." arXiv preprint arXiv:2106.04958 (2021).


Typos & Questions:
1. Algo. 1 gives a detailed algorithm procedure, but I can still be confused about obtaining the initial population/ policy pool? Is it a fixed policy pool, or would it be expanded by adding the new learned policy during the training?
2. Page 5: The repeat 'Environment' sub-section names of the last two paragraphs.

**Summary Of The Paper:**

This paper tries to find a new approach by enforcing the diversity in multi-agent RL  via the maximum entropy to address the zero-shot human-AI coordination problem. More specifically, in the proposed Maximum Entropy Population-based training (MEP) framework,  the authors choose population entropy as an efficient surrogate objective and use the prioritized sampling. The empirical results on the overcooked show that MEP outperforms other baselines with both simulated and real human players.


**Summary Of The Review:**

This paper presents an interesting solution to train the diversity policy with a population, but more clarifications (as stated above) are required.

---

> ### Author Response · Authors · 2021-11-12
> **To Reviewer 1grX**
>
> Thank you for your valuable feedback!
>
> - Yes, we agree that policy diversity is a general method that can be useful in many other problems, such as building robust AI in games and building a robot to complete tasks in the real world. We will discuss more about the broader impact of the paper.
>
> - The ablation study of prioritized sampling and entropy population is shown in Figure 5 (b) in the new version, where we use MEP_{α=0} to denote the MEP model without the population entropy reward, and MEP_{β=0} to denote the MEP model without the prioritized sampling mechanism, respectively.
>
> - Thank you for the references! We now add the discussion about the relationship between the proposed method and the diversity promoting solutions in the PSRO framework in the related work section of the paper: In games with non-transitive dynamics where strategic cycles exist, e.g., Rock-Paper-Scissors, Policy-Space Response Oracle (PSRO)-based methods (Balduzzi et al., 2019; Perez- Nieves et al., 2021; Liu et al., 2021) provides solutions to learn diverse behaviors. Our method is complementary to these previous works and could be combined with them.
>
> - The initial population/policy pool is obtained using the agents trained with the population entropy objective, see Algo.1, to encourage diversity.
>
> - In Page 5, the sub-section names of the last two paragraphs are “Environment” and “Experiments”, which are not repeated “Environment”.

---

### Official Review · Reviewer_5ykj · 2021-11-03

**Correctness:** 3
**Technical Novelty And Significance:** 3
**Empirical Novelty And Significance:** 2
**Recommendation:** 6
**Confidence:** 3

**Main Review:**


I am struggling to come up with sound overall grade for this paper. In short:

1. The method seems to be technically sound, building on simple and relatively popular ideas in both multi-agent and single-agent RL in an interesting way.

2. It seems to be effective when tested against comparable baselines in an interesting multi-agent environment.

3. The submission does a good job at providing enough details (and code) to enable quickly reproducing the work.

However:

4. The experimental section is relatively weak:

    a. Firstly, multi-agent Overcook is a relatively recent benchmark all things considered (and I would definitely not define it "popular"). There are a variety of choices, ranging from toy-like environments (matrix games, OpenAI's MPE) to complex ones (SMAC, Hanabi, DM Lab) that are popular in the recently wildly exploding MARL literature that could also have been employed, and possibly preferred. This would have enabled to understand the significance of the results compared to state of the art zero-shot coordination.

    b. This is particularly a problem when comparing this manuscript with Lupu et al., 2019, which the authors acknowledge being comparable work. Considering that they are comparing their methods against theirs, it seems unreasonable not to benchmark MEP on at least some of the toy problems proposed in that work (whilst I would accept the authors not trying to compete with their well-tuned Hanabi results, as that would result in quite a lot of potential work).

5. The focus on AI-human coordination seems a little weird. In principle this paper is proposing to increase the diversity of states seen by the agent policy by maximising the types of behaviours generated during training time. This means that the method is applicable to be tested against any kind of out-of-distribution(-ish) agents. This is not a huge issue, but implies that the experimental section could have focused more greatly on undestanding the space of ad-hoc policies over which it is more robust vs possible failure cases. In my opinion, generating an adversarial out of distribution set of agents to test against would have been more effective than qualitatively analysing a comparably small quantity of human trials.

6. The paper at times is unnecessarily handwavy, unclear, or makes dubious statements that are not well backed up by the literature. Here's some examples:

   a. In the introduction, it is stated that self-play is the "mainstream method for building state-of-the-art AI agents" (without reference), but self-play makes for a relatively limited amount of work when looking at the broad RL / MARL literature. I would argue that it is _barely_ mainstream, given the implementation complexity of even its simplest form. It is also stated that "self-play-trained agents are very specialized", but this is not a well understood property of PBT, and in practice doesn't seem to have significantly affected performance when properly done (see e.g. Alphastar).

   b. It is claimed that "prioritized sampling [of agents policies from the population] [makes] the collaboration between the AI agent and any agent in the population as good as possible in general" -- I don't understand how to interpret the sentence: is the manuscript saying that prioritized sampling is generally optimal (a fairly strong claim!) wrt. learning with PBT for cooperative MARL? How does this interact with the fact that the schema utilises a particular ranking system that might be more or less compatible with the task?

### Nits

- Generally: the [...] RL -> [...] RL
- Section 1: prioritized sampling [of what?]
- Section 1: the experimental section is not a contribution of the work -- there's nothing intrinsically new about how MEP was tested, as far as I can see?
- Section 2: modes of sub-optimal -> modes of optimal (?)
- Section 3.1: what do incentive and multi-modal mean here?
- Section 3.4: the first paragraph feels off in terms of syntax / punctuation.
- Figure 3: hard to interpret -- it feels like it could have been reduced to two plots by grouping wrt. y-axis and alpha.

**Summary Of The Paper:**

The authors propose a maximum entropy objective that can be applied to population-based multi-agent training in an RL setting to create a population of multi-agent policies that can more easily cope with zero-shot introductions of "unseen" policies. They formulate a framework for successfully applying this objective at scale by working out a proxy objective, and apply the method to the multi-agent Overcook environment tasks.

**Summary Of The Review:**

Overall, the manuscript presents an idea that seems compelling and useful, but the experimental section doesn't provide enough signal to compare this method against the literature. This makes my recommendation borderline at this point, so I'm looking forward to discussing the manuscript with the authors and the rest of the reviewers to understand how to improve it towards possibly acceptance.

---

Bumped up score to weak accept.

---

> ### Author Response · Authors · 2021-11-12
> **To Reviewer 5ykj**
>
> Thank you for your valuable feedback!
>
> - We newly added the comparison between MEP and TrajeDi on the matrix game in the updated version as you suggested.
> In the single-step collaborative matrix game, player 1 must select a row while player 2 chooses a column independently. Both agents get the reward associated with the intersection of their choices at the end of the game. We use the same evaluation protocol as proposed by Lupu et al 2021. From Figure 3 in the updated paper, we can see that MEP converges faster than TrajeDi. We did an extensive hyper-parameter search for TrajeDi, as shown in Figure 7 in Appendix F.
>
> - Yes, the algorithm is not limited to human-AI coordination. The proposed MEP method is also applicable in other out-of-distribution scenarios. We leave the exploration of MEP in other out-of-distribution sceneries to future work.
>
> - In the introduction, we did cite Alphastar for the claim that “self-play is the mainstream method for building state-of-the-art AI agents.” About the claim "self-play-trained agents are very specialized,” we cite the work by Carroll et al., 2019 in the second paragraph of the introduction. Carroll et al., 2019 addressed that self-play-trained agents are very specialized in their experiments. To be more specific, for example, in the Overcooked game, the self-play-trained agents only use a specific pot and ignore the others. However, humans use all pots. The AI agent ends up waiting unproductively for the human to deliver a soup from the specific pot, while the human has instead decided to fill up the other pots.
>
> - About the claim “prioritized sampling of agents policies from the population makes the collaboration between the AI agent and any agent in the population as good as possible in general,” we mean that with prioritized sampling, the AI agent not only learns to collaborate with the easy-to-cooperate ones but also learns to cooperate with the harder-to-cooperate ones. We did not mean that prioritized sampling is generally optimal. We now changed this sentence in the updated version.
>
> - We now fixed the grammar nits in the updated version.

---

> > ### Comment · Reviewer_5ykj · 2021-11-29
> > **Response**
> >
> > Thank you for your rebuttal, and apologies for the delay writing a response.
> >
> > 1. Thank you for incorporating the experiments on the matrix games. I think they are very helpful in indicating that some of the hypotheses made in the manuscript regarding MEP are indeed likely to be sound. Also fairly impressed that your implementation of TraJeDi largely matches the original paper's, so well done!
> >
> > 2. I do think this choice makes the experimental section a little weaker than it should be. To me it seems like quite the missed opportunity to understand both qualitatively and quantitatively policies produced by MEP.
> >
> > 3. Two papers (one of which is practically impossible to reproduce in any meaningful way at this point in time) do not make for a "mainstream" technique. I believe my point stands.
> >
> > > we mean that with prioritized sampling, the AI agent not only learns to collaborate with the easy-to-cooperate ones but also learns to cooperate with the harder-to-cooperate ones
> >
> > 4. Right, but why is that the case? Looking at you experiments, it is clear that prioritising samples using the selected bias does seem to empirically improve learning in your settings, but it is not enough to just state it as a general fact. It is otherwise a largely circular argument.
> >
> >
> > Overall, I am not sure I am ready to completely recommend the paper as is for acceptance, as I share some concerns regarding the comparison between MEP and TraJeDi as highlighted by reviewer mavT. I don't think it will be possible to completely compare the two algorithms without matching the experimental settings appropriately.
> >
> > That said, it does seem like MEP is overall a good +1 over TraJeDi in terms of algorithmic ergonomics, Hanabi is not an easy environment to deal with (and certainly too hard to add in the course of this reviewing period), and the authors have put it effort to add additional experiments, so I think there's room to bump up my score to a weak accept.

---

### Official Review · Reviewer_PRq2 · 2021-11-03

**Correctness:** 3
**Technical Novelty And Significance:** 3
**Empirical Novelty And Significance:** 3
**Recommendation:** 3
**Confidence:** 3

**Main Review:**

Strengths
The problem is important, and testing it on Overcooked along with a human experiment is a good domain for this exploration.
The population diversity metric is also reasonable, and seems like a good quantity to include as part of an auxiliary loss.
The human experiments are neat (although I am left wondering why TrajeDi is not compared with in the human experiments).


Weaknesses

Though the paper shows the population entropy to be a lower bound of the population diversity, the gap is huge (factor of n^2). Looking at the entropy values in Table 1, the lowest value is about 0.9 and the highest is about 1.8 (only a factor of 2). As such, even for the smallest choice of n=2, it seems that the lower bound ends up being nearly useless. I'm surprised the paper actually proposes to train based on the lower bound. To me, optimizing an approximation of the population diversity seems much more promising.

I also thought section 3.4 was not well motivated, and it is unclear what the takeaway of this section is. On one hand, it's unclear why we are interested in optimizing the worst performing partner, rather than the average reward of the partners. Moreover, looking at the supporting claims for Eq 10 (Lemma 4), I don't have intuition as to why this claim is not completely trivial -- that training with the lowest performing agent improves the minimum reward over all agents (which seems obvious). Related to this, why does the paper claim uniform sampling "does not provide any guarantees" (can't we also claim that training with all agents uniformly increases the overall reward?)

I'm also not convinced by the discussion in Q1 regarding Figure 3. It does not appear that the entropy is converging to a higher value. In fact the plots are very spiky and hard to interpret. Judging from Table 1, however, it does appear that pushing for population entropy does impact the environment reward quite a bit, contrary to the claims in the paper that "the reward does not decrease much".

I found the writing to be poor. There are many choppy parts that break the reader's rhythm such as "less 'panic'", "While uniform sampling does not provide any guarantee on the worst case.". There are also very vague statements, such as "With prioritized sampling, we make the collaboration between the AI agent and any agent in the population as good as possible in general". It's unclear to me what precise claim they are making. Of course there are no guarantees regarding the global optimum, so what precisely does it mean to "make... as good as possible in general"?

**Summary Of The Paper:**

The paper aims to address the important problem of training AI agents in multi-agent games. Standard self-play training leads to agents that overfit to a particular partner, and even naive population based training may not help much if the population is not very diverse. A number of recent works have looked into generating diversity in the partner strategies so that population based training methods can work better. By covering a diverse set of trajectories/policies, it is more likely that coordinating with a new partner (perhaps a human) at test time will fall in-distribution relative to our training set partners.

The paper proposes a population diversity metric based on cross-entropy between different pairs of agents in the population. They optimize a lower bound to this metric based on the population entropy. They also propose a 'prioritized sampling' procedure that determines which partner in the population the ego-agent should train with. They experiment on the game of Overcooked, compare with TrajeDi, and run a human experiment to check performance of coordination with humans at test time.

**Summary Of The Review:**

The paper tackles an important problem, but leaves much room for improvement. The population entropy should be backed up by (toy) experiments that analyze the effect of the quantity, since on paper it seems that the lower bound exhibits a huge gap compared to the empirical range of entropy values observed in Table 1. The experiments can also be improved (TrajeDi is missing in the human experiments), and the current results are hard to interpret (the reward in Table 1 does drop significantly when the entropy term is pushed up).   Improving the overall writing would also raise the paper's potential impact.

---

> ### Author Response · Authors · 2021-11-12
> **To Reviewer PRq2**
>
> We thank the reviewer for the detailed comments.
>
> - About the n^2 in Theorem 1, you may have a **misunderstanding**. The “n” represents the population size, which is a constant during training. The factor of n^2 is also a constant. Therefore, the lower bound of population diversity is proportional to population entropy. There is no n^2 gap between the population diversity and the population entropy objective.
>
> - Section 3.4 introduces the prioritized sampling for training the cooperative agents, which assigns a higher priority for the agents that are harder to cooperate with. Since we do not know how a human player would perform in reality, we are interested in optimizing the worst-performing partner to provide a guarantee on the non-optimal cases. In the case that the human player is not very good at the game, our agent can still be cooperative and achieve a relatively high score. If we use uniform sampling, the AI agent could not learn how to cooperative well with the hard-to-cooperate partners; it only learns to cooperate with the easy-to-cooperate partners. When the human player is hard to cooperate with, the total performance degrades significantly.
>
> - Regarding Figure 3 and Q1, the entropy indeed coverages to a higher value regarding the best-performing policy, 1.12 vs. 1.139, which are also shown in Table "Best self-play reward and its corresponding population entropy with different α" in Appendix in the new version. This table shows the hyper-parameter trails about the entropy bonus weight. There is a trade-off between performance and diversity. If the weight of the entropy is too high, it might hurt the performance. If we tune the weight carefully, we can find a weight that does not make the reward decrease much.
>
> - "With prioritized sampling, we make the collaboration between the AI agent and any agent in the population as good as possible in general.” We originally mean that the AI agent can not only cooperate with the easy-to-cooperate partners but also cooperate relatively well with the harder-to-cooperate partners. We now changed this sentence in the updated paper.
>
> - The effect of the population entropy is analyzed in Table 1, Figure 4, Table 2 in Appendix of the new paper, and the supplementary video. We will try to include more analysis on the population entropy. The gap of the lower bound is clarified in the first bullet point. We didn't do the TrajeDi experiments in the human experiments because the experiments with the actual human are rather expensive and time-consuming. The cost exceeds our budget. Besides, the experimental results with the human proxy model are similar with the real-human experiments. The final experiment results are shown in Figure 5 and Figure 6.

---

> > ### Comment · Reviewer_PRq2 · 2021-11-24
> > **Thanks for response**
> >
> > Thanks for the response. I have also read the other reviewers' comments. I don't have any further questions, but I am still not that excited by this work. To me the contribution that the paper is trying to claim can be clearly stated as -- "using a surrogate objective that is different from TrajeDi by adding a term for individual policy entropy".
> >
> > With this in mind, the shortcomings of the paper are 1. the limited novelty and 2. how the paper spends unnecessary time discussing population diversity, prioritized sampling, and not enough empirical exploration to convince the reader. Given the limited novelty / technical advancements, much more time should be spent on experimental validation (baselines, hyperparameters, as other reviewers suggest), ideally more convincing experiments as to why the extra individual entropy term is SO important to be worth presenting to the community.
> >
> > -------
> > -------
> >
> > > About the n^2 in Theorem 1, you may have a misunderstanding. The “n” represents the population size, which is a constant during training. The factor of n^2 is also a constant. Therefore, the lower bound of population diversity is proportional to population entropy. There is no n^2 gap between the population diversity and the population entropy objective.
> >
> > Yes, I understand that n is a constant during training (it seems like the paper chooses n=5). My concern is that the optimization landscape for population entropy may have little to no resemblance to the optimization landscape for population diversity (e.g. even though it is a lower bound, there is a factor of 25 gap). As such, I find the exposition in section 3.1 / 3.2 trying to link the two quantities a little odd.
> >
> > So, then ignoring population diversity for now, the paper proposes a surrogate objective of maximizing entropy of the "mean" policy. As the discussion with other reviewers point out, this is similar to the TrajeDi objective, except there is an extra term for the individual entropies.
> >
> > > Since we do not know how a human player would perform in reality, we are interested in optimizing the worst-performing partner to provide a guarantee on the non-optimal cases.
> >
> > Yes, this was my understanding during my initial review. My confusion was in why we would be interested in optimizing for such a metric, instead of just optimizing for the average reward. But this point is not so important, so there's no need to discuss further.

---

### Author Response · Authors · 2021-11-12
**Revision**

Thank the reviewers for the valuable feedback!
We updated the paper with new experiments on the matrix game environment and also fix some issues addressed by the reviewers.
Video link: https://youtu.be/Xh-FKD0AAKE

---

### Decision · Program_Chairs · 2022-01-20

**Decision:**

Reject

**Comment:**

This paper has several issues:
(1) The empirical results were incomplete and hard to interpret.
1.a The paper uses non-standard benchmark domains making comparisons with results in the literature very difficult. The paper does not use the same environments as related baselines they build on. Some progress on this last point was made during the discussion period---well done authors!
1.b The experiments did not sweep key hyperparameters of the TrajeDi baseline, and generally did not comment on nor ablate several other potentially important hyperparameters and design choices
(2) the proposed method is very similar to another called TrajeDi and the paper and results don't clearly show why the modification of TrajeDi is significant (see #1). The paper initially claimed the TrajeDi was a concurrent submission but one reviewer pointed out the work was published in May
(3) writing and structure could be improved. In addition some inaccurate statements could be cleaned up
(4) The algorithm is more generally applicable beyond human-AI coordination and the reviewers found it odd the paper did not focus on this

In addition, the authors did not respond to several of the reviewers responses. This made it difficult for the reviewers to increase their scores. Several reviewers found the work intriguing, but its not ready yet